# Topical application of an irreversible small molecule inhibitor of lysyl oxidases ameliorates skin scarring and fibrosis

Nutan Chaudhari[1,2], Alison D. Findlay[3], Andrew W. Stevenson [1],
Tristan D. Clemons[2,4], Yimin Yao[3], Amar Joshi[3], Sepidar Sayyar[5,6],
Gordon Wallace [5,6], Suzanne Rea[7], Priyanka Toshniwal[1], Zhenjun Deng[1],
Philip E. Melton[8,9,10], Nicole Hortin[1], K. Swaminathan Iyer[2], Wolfgang Jarolimek[3],
Fiona M. Wood[1,7] & Mark W. Fear [1] ✉

Scarring is a lifelong consequence of skin injury, with scar stiffness and poor appearance presenting physical and psychological barriers to a return to normal life. Lysyl oxidases are a family of enzymes that play a critical role in scar formation and maintenance. Lysyl oxidases stabilize the main component of scar tissue, collagen, and drive scar stiffness and appearance. Here we describe the development and characterisation of an irreversible lysyl oxidase inhibitor, PXS-6302. PXS-6302 is ideally suited for skin treatment, readily penetrating the skin when applied as a cream and abolishing lysyl oxidase activity. In murine models of injury and fibrosis, topical application reduces collagen deposition and cross-linking. Topical application of PXS-6302 after injury also significantly improves scar appearance without reducing tissue strength in porcine injury models. PXS-6302 therefore represents a promising therapeutic to ameliorate scar formation, with potentially broader applications in other fibrotic diseases.

Currently, there are no effective therapeutics to ameliorate scar formation and, consequently, patients with significant injuries often endure repeated and expensive surgical and adjunct interventions to improve the scar. Scar appearance and stiffness are largely a result of excess, densely packed collagen in the dermal extracellular matrix (ECM), a consequence of wound repair.

The lysyl oxidase family of enzymes [comprising 5 members; lysyl oxidase (LOX) and lysyl oxidase-like 1-4 (LOXL1-4)] plays a critical role in collagen deposition and stability. Their primary function is the oxidation of side chain lysine residues in collagen and elastin, leading to the formation of aldehydes which spontaneously react to form covalent crosslinks. Initially, divalent crosslinks are formed, also known as the reducible or immature crosslinks dehydrohydroxy-lysinonorleucine (HLNL) and dehydrodihydroxy-lysinonorleucine (DHLNL)[1]. HLNL and DHLNL undergo further reactions to form stable, trivalent compounds also called mature or non-reducible crosslinks such as pyridinoline (PYD) and deoxypyridinoline (DPD)[2]. Under normal physiological conditions, crosslinks are essential for the

[1]Burn Injury Research Unit, School of Biomedical Sciences, The University of Western Australia, Crawley, Australia. [2]School of Molecular Sciences, The University of Western Australia, Crawley, Australia. [3]Drug Discovery Department, Pharmaxis Ltd, Sydney, NSW, Australia. [4]School of Polymer Science and Engineering, University of Southern Mississippi, Hattiesburg, MS 39406, USA. [5]Intelligent Polymer Research Institute, Australian Research Council Centre of Excellence for Electromaterials Science, University of Wollongong, Wollongong, NSW 2500, Australia. [6]Australian National Fabrication Facility—Materials Node, Innovation Campus, University of Wollongong, Wollongong, NSW 2500, Australia. [7]Burns Service of Western Australia, WA, Department of Health, Nedlands, WA, Australia. [8]Menzies Institute for Medical Research, University of Tasmania, Hobart, TAS, Australia. [9]School of Pharmacy and Biomedical Sciences, Curtin University, Bentley, WA, Australia. [10]School of Global Population Health, University of Western Australia, Crawley, Australia. ✉e-mail: mark.fear@uwa.edu.au

structural integrity of collagen, elastin, and the skin[3]. However, under some pathological/ disease settings, including scar formation, increased production of ECM proteins leads to excessive mature cross-link formation, increased matrix stability and, ultimately, fibrosis[4–6].

β-aminoproprionitrile (BAPN) is a well-known inhibitor of all lysyl oxidases (pan-LOX inhibitor). When given systemically to developing animals, BAPN results in lathyrism (weakness and fragility of connective tissues such as skin, bones and blood vessels as a result of increased collagen solubility)[7–10]. Nevertheless, the fundamental mechanism by which lysyl oxidase inhibition diminishes, and potentially reverses, fibrosis has warranted evaluation in clinical studies[10]. In patients with scleroderma[9,10] long-term BAPN treatment increased the relative amount of soluble collagen, providing a mechanistic proof of concept. Shorter-term treatment has also proven effective in preventing the recurrence of keloids after surgery[10,11]. Prolonged administration of BAPN does, however, result in toxicity[12], including allergic skin rash, anemia[9,12,13], and lathyritic bone changes[12,14]. These side effects are dose-related and likely driven by off-target and metabolite-mediated effects. Inhibition of LOX and LOXL enzymes with a more selective compound designed for topical application therefore remains a promising approach to reduce collagen deposition in scarring and fibrosis. Here we show that fluoroallylamine-based small molecule pan-LOX inhibitors readily permeate skin when applied topically, and reduce collagen deposition and crosslinking in vitro. The inhibitors reduce skin fibrosis and improve scar appearance in animal models, suggesting they may be effective in the amelioration of scar formation after skin injury.

## Results

### PXS-4787 is a mechanism-based specific and effective pan-LOX inhibitor

BAPN has long been recognized as an irreversible, mechanism-based small molecule inhibitor of all lysyl oxidases, yet its shortcomings have limited widespread clinical use. Based on the demonstrated successes of fluoroallylamine-based inhibitors for semicarbazide-sensitive amine oxidase (SSAO) and LOXL2[15,16], we sought to design a series of pan-lysyl oxidase inhibitors incorporating this motif. At the outset, key drivers and measures of success were deemed to be comparable potency for all lysyl oxidase isoforms, an irreversible mode of inhibition and good selectivity over related amine oxidases. Moreover, in light of the toxicities associated with BAPN, postulated to be a consequence of unwanted substrate activity at several amine oxidases, including SSAO and diamine oxidase (DAO)[14], identifying a compound devoid of off-target substrate capacity was an important criterion. Lastly, as the intended route of administration was topical, a development candidate with a small molecular weight and excellent permeability was sought.

Using the endogenous substrate lysine as inspiration and extrapolating to mofegiline (reported in patent literature to be a modest inhibitor of chicken lysyl oxidase activity[17]) to incorporate the fluoroallylamine necessary for mechanism-based inhibition, provided a sound starting point from which to begin structure-activity-relationship (SAR) investigations (Fig. 1a, b). To more readily facilitate analog preparation an oxygen linker was incorporated, providing 1 and 2. However, as this modification resulted in a slight reduction in lysyl oxidase (LOX) potency we took the opportunity to survey alternative fluoroallylamine configurations (including 3) before identifying 4 as a promising candidate (LOX IC$_{50}$ 8 μM) albeit with substantial SSAO substrate activity. To address this shortcoming we investigated the nature of the linker and, while introduction of nitrogen (5 and 6) and sulfur[7] proved unhelpful, incorporation of a sulfone, resulting in PXS-4787, provided a compound with the desired level of balanced lysyl oxidase activity and no unwanted off-target amine oxidase activity (as either inhibitor or substrate).

The irreversible nature of PXS-4787-mediated inhibition was confirmed by a jump dilution assay (Fig. 1c) in which 100-fold

dilution from 10x the IC$_{50}$ led to only a small (9%) recovery in LOXL1 activity (LOXL1 was used as a surrogate for LOX owing to similar pharmacology). Intrigued by the stark difference in (irreversible) inhibition displayed by PXS-4787 (SO$_2$-linker) compared to 4 (O-linker) we also profiled the corresponding sulfone analog of 3 (denoted 8 in Fig. 1c) and a similar trend was observed. Overall, the sulfone linker in PXS-4787 is both unique and critically important in achieving a compound with a suitable profile for successful clinical development.

PXS-4787 dose-dependently inhibits lysyl oxidases with IC$_{50}$ values ranging from 0.2 μM (LOXL4) to 3 μM (LOXL1) (Fig. 2a) and displays comparable inhibitory activity across species (Supplementary Table S1). As expected for a mechanism-based inhibitor, PXS-4787 displays increased potency upon longer pre-incubation times (Fig. 2b). In jump dilution experiments, PXS-4787 demonstrated irreversible inhibition of LOXL1, LOXL2, and LOXL3 (3, 9, and 8% recovery of enzyme activity, respectively, Fig. 2c). In contrast, the activity of enzyme exposed to a reversible inhibitor (lacking the fluoro leaving group) was virtually completely recovered. PXS-4787 competes with a model substrate (putrescine) for the LOX binding site (Fig. 2d). The allylamine of PXS-4787 interacts with the co-factor of the enzyme and elicits mechanism-based inhibition (Fig. 2e). Through careful optimization, and in contrast to BAPN, PXS-4787 selectivity targets lysyl oxidases and is neither an inhibitor of, nor a substrate for, related oxidases (Fig. 2f, g and Supplementary Table S1). Furthermore, PXS-4787 was clean when profiled in a broad panel of macromolecular targets using enzyme and radioligand binding assays (LeadProfilingScreen, Eurofins CEREP Panlabs, 68 assays, test concentration 10 μM; Supplementary Data S1).

### LOX expression in normal skin and scar tissue and fibroblasts

Injury is known to cause an upregulation of lysyl oxidases. To determine whether these changes are sustained over time, qPCR for *LOX* and *LOXL* transcripts was conducted using fibroblasts isolated from matched scar and normal skin samples. *LOX* and *LOXL1* were significantly increased in scar fibroblasts compared to normal skin (Fig. 3a). Immunohistochemistry for LOX was also performed from the tissue samples of normal human skin tissue and normotrophic scar tissue from which fibroblasts were isolated at least 2 years after injury. LOX protein expression was observed in epithelial and endothelial cells as well as dermal cell populations and significantly increased in scar when compared to the matched control skin (Fig. 3b–d).

### PXS-4787 reduces deposition and crosslinking of collagen I secreted by human fibroblasts in vitro

To establish that PXS-4787 does not have any unwanted effects on cell viability, primary human fibroblasts were treated with a range of PXS-4787 concentrations (0–100 μM) for 72 h. PXS-4787 had no effect on fibroblast cell viability (Supplementary Fig. S1) and treatment of HepG2 cells with the same range of concentrations did not induce phospholipidosis (score 0, data not shown). In pharmacological experiments, the highest concentration used was 10 μM which achieved complete inhibition of the least sensitive lysyl oxidase after 2 h.

To determine the effects of PXS-4787 on collagen cross-link formation, primary human fibroblasts from six different participants (six biological replicates) were cultured for 11 days using 'scar-in-a-jar' conditions in the presence of 0, 1, and 10 μM of inhibitor. Liquid chromatography with tandem mass spectrometry (LC-MS/MS) was used to measure hydroxyproline and cross-link concentrations in these samples. Hydroxyproline concentration provides a close estimate of collagen concentration as it is present in high concentrations in collagens and is critical to fibril formation. Baseline levels of hydroxyproline and crosslinks showed high variation between biological samples, likely reflecting differences in age, gender, and body site

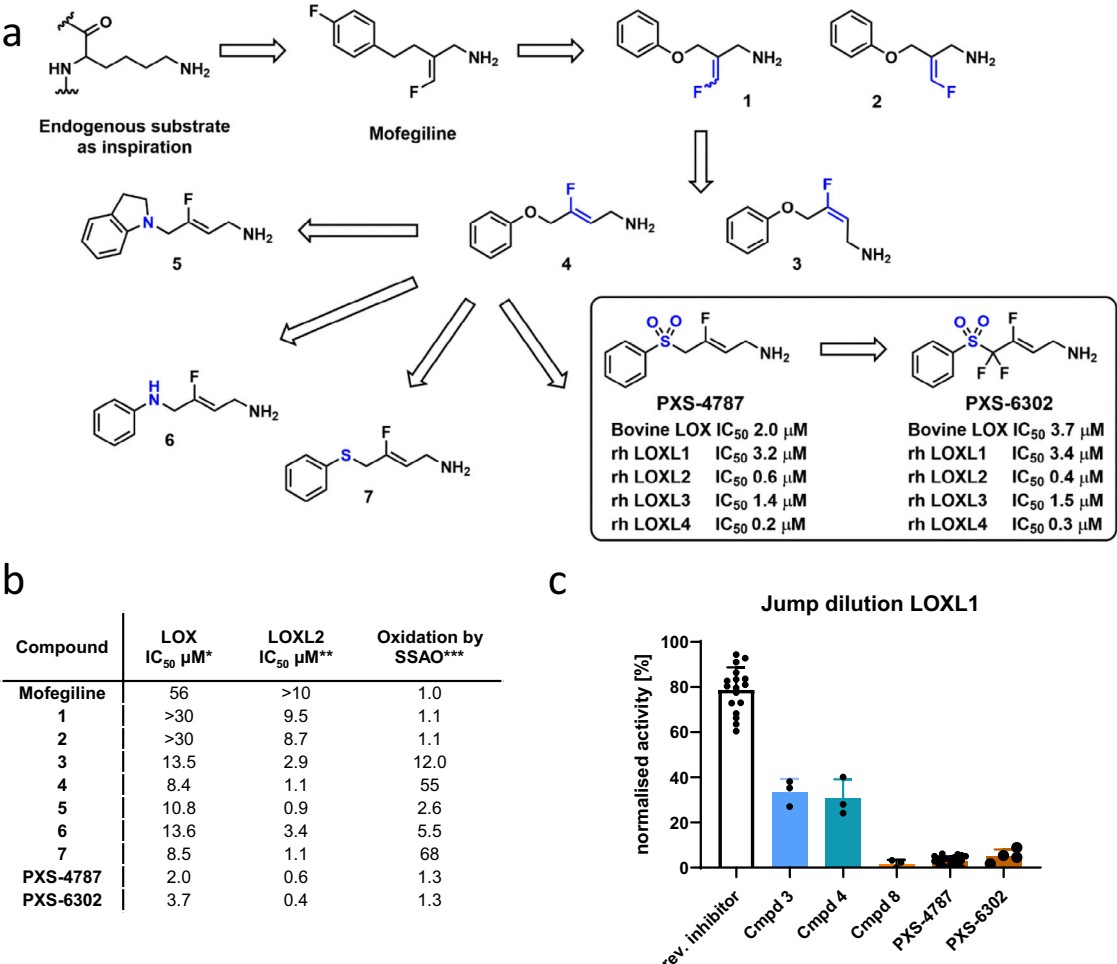

**Fig. 1 | Discovery and characterization of pan-lysyl oxidase inhibitors, PXS-4787 and PXS-6302. a** Analog generation en route to the discovery of PXS-4787 and PXS-6302. **b** The measurement of enzymatic inhibitory activity was measured using an Amplex Red oxidation assay, as described previously[16]. For compound oxidation assays, the amount of hydrogen peroxide generated at a single concentration (30 μM) of compound relative to DMSO was measured. Minimum of $n = 2$ for each experiment. *Bovine (aorta) LOX used for analog screening due to ready availability and pharmacological similarity to native human LOX. **Human recombinant LOXL2. ***Human recombinant SSAO. **c** Jump dilution assay used to measure the irreversible nature of LOXL1 inhibition (LOXL1 was used as a surrogate for LOX owing to similar pharmacology (data are presented as mean ± standard deviation ($n = 17$ independent experiments for reversible inhibitor, $n = 3$ for cpd 3, 4 and 8 and $n = 22$ and 4 for PXS-4787 and PXS-6302 respectively)). Source data are provided as a Source Data file.

from which cells were isolated. At a concentration of 1 μM, PXS-4787 did not reduce the hydroxyproline concentration. However, 10 μM significantly (p = 0.0385) diminished hydroxyproline concentration, albeit to a small degree when compared to control (Fig. 4a). The concentration of DHLNL, the more abundant immature cross-link, was dose-dependently reduced by PXS-4787 (Fig. 4b) while HLNL showed a trend reduction at the highest tested concentration (Fig. 4c). Both types of mature crosslinks (DPD and PYD) were dose-dependently reduced (Fig. 4d, e). Interestingly, a significant and sustained reduction in collagen I production was observed after PXS-4787 treatment at the highest dose (Fig. 4i).

Having demonstrated PXS-4787 treatment significantly reduces cross-link formation and collagen secretion in de novo matrix production in vitro, we next examined whether this reduction in crosslinks alters the ECM structure. Fibrillar collagen I deposition was measured using confocal image analysis (Fig. 4f–k). Deposited fibrillar collagen I was significantly reduced in the 10 μM treatment group (p < 0.01, Fig. 4f) compared to control and 1 μM treatment. Furthermore, coherency analysis[18] was used to determine collagen alignment. This analysis specifically analyses the alignment of collagen fibers, with more alignment increasing the score and more random distribution provided a lower score. In scar tissue scores are higher as fibers are

densely parallel aligned in contrast to normal skin[18]. In both treatment groups PXS-4787 significantly reduced coherency (p < 0.01) when compared to control (Fig. 4k) suggesting inhibition of crosslinking reduces the stability of deposited collagen and changes the distribution to be more like that seen in normal skin in vitro. No change in *COL1A1* or *LOX* RNA levels was observed (Supplementary Fig. S4) indicating treatment does not impact *COL1A1* or *LOX* transcription.

Other studies have suggested LOX has non-enzymatic functions linked to epithelial differentiation[19,20]. Given that PXS-4787 would be applied topically, we investigated whether PXS-4787 had an impact on both keratinocyte and fibroblast transcriptomes using RNASeq. Since the target of interest is extracellular LOX activity, we wanted to understand if PXS-4787 also affected intracellular LOX activity. PXS-4787 (10 μM) was applied to cultured fibroblasts and keratinocytes isolated from five different patients and treated for 24 h. RNASeq analysis showed only four genes with significant differential expression (FDR < 0.05) in fibroblasts and only two differentially expressed genes in keratinocytes (Supplementary Fig. S2 and Supplementary Table S3). The underlying biological mechanisms related to extracellular matrix biology of these six genes are unclear and these results suggest PXS-4787 does not impact on intracellular functions of LOX relevant for skin biology. This suggests application of the compound should

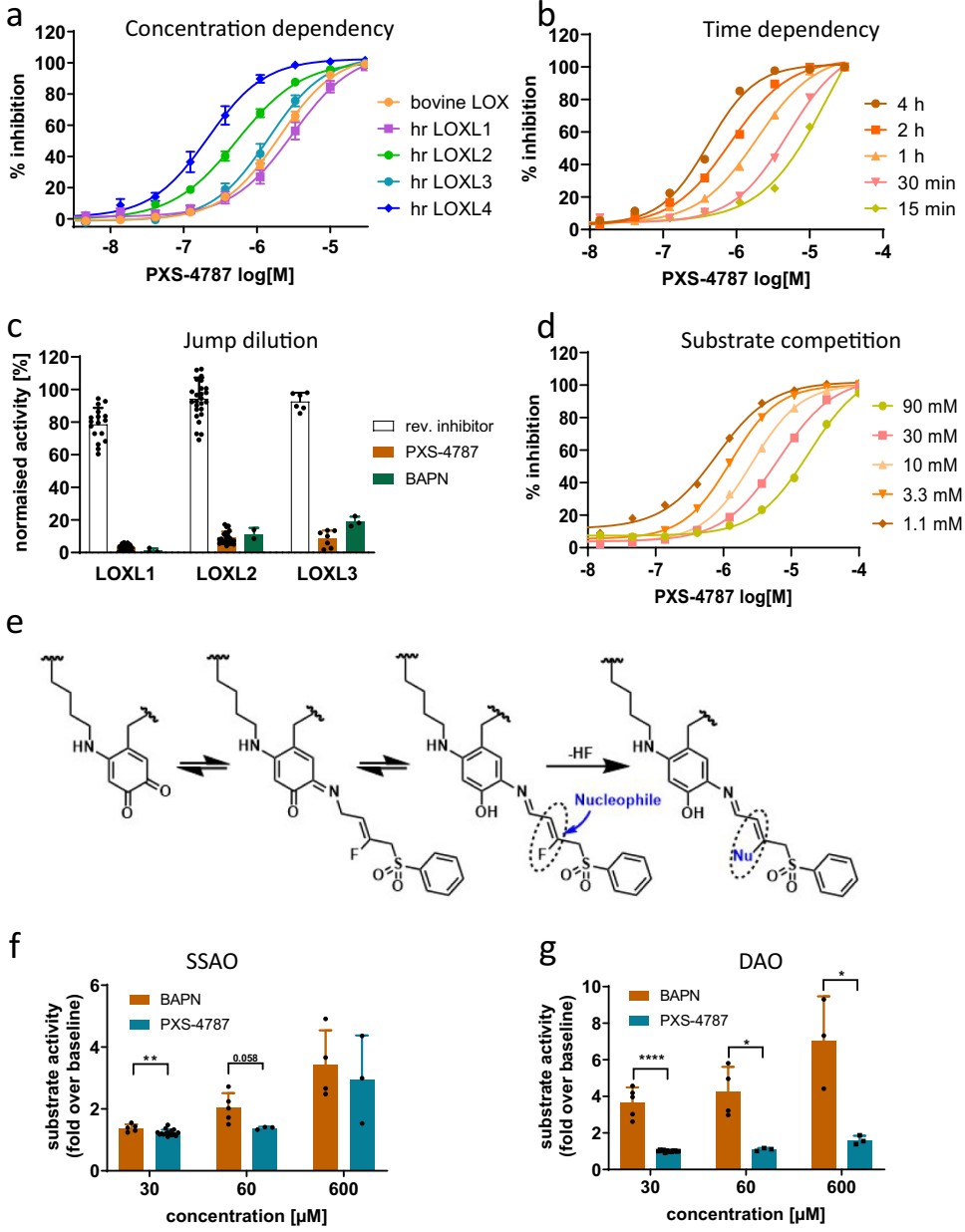

**Fig. 2 | Characterization of the mode of action of PXS-4787. a** Lysyl oxidase inhibitory profile of PXS-4787 ($n = 11$). **b** Time-dependent increase in potency of PXS-4787 ($n = 12$ at each timepoint). **c** Irreversible inhibition of LOXL1, LOXL2, and LOXL3 by PXS-4787 and BAPN as measured in a jump dilution assay and compared to a reversible inhibitor. **d** Substrate (putrescine) competition showing a reduction of potency with increasing substrate concentration due to competition for the enzymatic pocket. **e** Postulated mechanism of irreversible inhibition by PXS-4787. **f** Oxidation of PXS-4787 and BAPN by SSAO ($n = 2$–4 independent experiments, $p$ values 0.0097, 0.058 for 30 and 60 micromolar concentration, respectively). **g** Oxidation of PXS-4787 and BAPN by DAO ($n = 2$–4 independent experiments, $p$ values $< 0.0001$, 0.012 and 0.019 for 30, 600 micromolar concentration respectively). Data are expressed as mean ± SEM. Subsequent statistical analysis was performed with unpaired two-sided Student's $t$ tests. $*p < 0.05$, $**p < 0.01$, $***p < 0.001$ and $****p < 0.0001$. Source data are provided as a Source Data file.

effectively target extracellular LOX without impacting on skin cell phenotype or leading to a compensatory response.

## PXS-6302, a second-generation analog with improved drug-like properties

PXS-4787 proved that the shortfalls of the archetypal pan-lysyl oxidase inhibitor BAPN can be overcome, achieving an excellent efficacy and selectivity profile. Stability data in cream formulations indicated that some unwanted degradation occurred which stopped further preclinical development. This necessitated the design of an analog, PXS-6302 (Fig. 1a), chemically modified to improve stability in cream formulations, that operates via the same irreversible mechanism of

lysyl oxidase inhibition as PXS-4787. Like PXS-4787, PXS-6302 is a more selective inhibitor than BAPN and is also optimally designed for topical application (for in vitro potency, selectivity and off-target screening data see Supplementary Table S1, Supplementary Data S2). It is a small (molecular weight < 300), hydrophilic molecule with high permeability across artificial membranes, a pre-requisite for good skin penetration, as determined by in vitro and ex vivo measurements. In the parallel artificial membrane permeability assay (PAMPA) assay, PXS-6302 demonstrated high permeability. The penetration of PXS-6302 across a monolayer of cells was tested using Caco-2 or MDCKII cells transfected with P-glycoprotein (Pgp). In both assays PXS-6302 exhibited a high permeability that was unaffected by the presence of

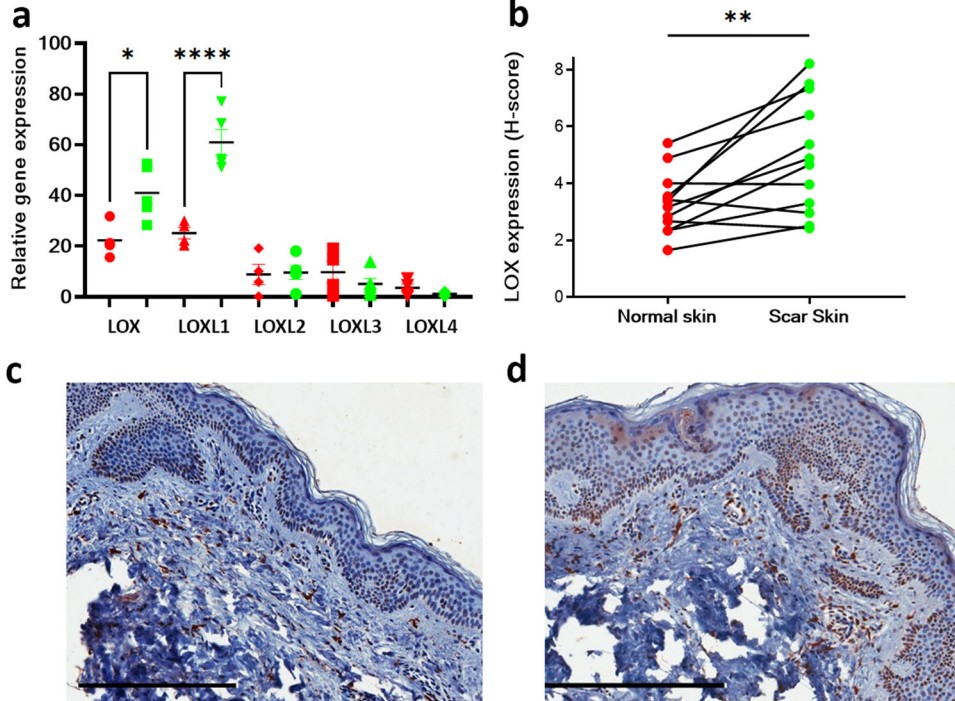

**Fig. 3 | Abundance of LOX/LOXL enzymes in human skin dermis and scar.** qPCR of LOX family of enzymes from matched normal skin and scar fibroblasts (**a**). LOX and LOXL1 expression is significantly increased in scar fibroblasts ($p = 0.0317$ and $p < 0.0001$ respectively using Mann–whitney two-tailed test). **b** Immunohistochemistry showed increased LOX intensity in scar tissue samples when compared to normal skin ($n = 11$ matched biologically independent human samples of scar and skin). Images of skin (**c**) and

normotrophic scar (**d**) immunohistochemistry for LOX from an individual patient ($n = 11$ independent biological pairs were imaged and this shows one matched pair of normal skin and scar from a single individual). Data are shown for each matched pair of normal skin and scar samples. Subsequent statistical analysis was performed using Wilcoxon matched-pairs signed rank test ($p = 0.0068$). Scale bars 300 microns. $*p < 0.05$, $**p < 0.01$, $***p < 0.001$ and $****p < 0.0001$. Source data are provided as a Source Data file.

the Pgp-inhibitor (Supplementary Table S2). After application of 3% cream to the epidermis of human skin ex vivo, the concentration of PXS-6302 in the opposite reservoir chamber increased 10-fold between 2 and 6 h, with a further 10-fold increase over the next 14 h, demonstrating good skin penetration similar to that observed for PXS-4787 (Supplementary Fig. S4).

The ability of PXS-6302 to engage the desired target in vivo (i.e. effectively inhibit lysyl oxidase activity in the skin) was tested in a rat pharmacokinetic-pharmacodynamic model. PXS-6302 was formulated as an oil in water cream of different concentrations (0, 0.3, 1, 10%) and applied to a shaved area on the back of a rat (500 mg cream applied to 16 cm²). After 24 h animals were sacrificed, the concentration of PXS-6302 in the skin was measured and the lysyl oxidase activity determined. The skin concentration of PXS-6302 was found to dose-dependently increase (0%: 0; 0.3% $7.9 \pm 1.5$; 1%: $8.9 \pm 1.4$%; 10%: $119 \pm 7.8$ µg/g; all $n = 3$), resulting in a corresponding reduction in lysyl oxidase activity. To quantify the degree of lysyl oxidase inhibition, a signal over noise ratio was calculated for vehicle and drug-treated skin biopsies and showed a dose-dependent inhibition of lysyl oxidase activity. PXS-6302 cream at a concentration of 1% achieved a 67% inhibition of the specific signal after 24 h, while 10% cream completely abolished the signal (98% inhibition) (Fig. 5a). Recovery of the activity in the healthy skin was slow; in animals that were sacrificed 48 h after the last dose (1% PXS-6302) activity was 26% of pre-drug activity.

To investigate the effect of repeat dosing, cream at different concentrations (0, 0.5, 1, and 3% PXS-6302, 400 mg applied to 16 cm²) was applied once daily for 5 consecutive days and 24 h after the last dose, animals were sacrificed and the skin concentration determined. PXS-6302 concentration in the skin again increased in a dose-dependent manner (0%: 0; 0.5%: $4.9 \pm 0.8$; 1%: $9.3 \pm 0.7$; 3%: $31.7 \pm 3.28$ µg/g; all $n = 4$), with the concentration measured after

repeated dosing of 400 mg of 1% cream for 5 days similar to that measured 24 h following a single dose (500 mg of 1% cream) suggesting that there was no large accumulation of drug over time. Lysyl oxidase activity was strongly reduced after 5 days (Fig. 5b).

**Topical application of PXS-6302 reduces skin fibrosis in repeat dose bleomycin-induced skin fibrosis in mice**

With a pharmacokinetic and pharmacodynamic profile supportive of the potential as a potent topical anti-fibrotic treatment, we next moved to evaluate the in vivo efficacy of PXS-6302. Intradermal injections of bleomycin on alternate days for three weeks induces fibrosis[21,22]. Exploratory studies involving the topical application of the first-generation compound PXS-4787 cream showed that the drug reduced fibrosis in the extracellular matrix, pharmacologically demonstrating the significant role of lysyl oxidases in bleomycin-induced skin fibrosis (Supplementary Fig. S5). In this model, starting from Day 3 and continuing until Day 20, bleomycin-dosed mice (0.1U, alternate days), were treated once daily topically with 0% (vehicle) or 1.5% PXS-6302 oil in water cream (2 ×100 µL applications at a 10-min interval). Topical treatment with 1.5% PXS-6302 resulted in a significant reduction in LOX activity (Fig. 6a), as well as reduced hydroxyproline content (Fig. 6b) and immature crosslinking (DHLNL and HLNL, Fig. 6c, d). Fibrosis was significantly reduced as assessed by skin thickness (Fig. 6e), and composite skin score, including assessment of Masson trichrome stained sections (Fig. 6f, Supplementary Fig. S6), with immunohistochemistry for COL1 and LOX both significantly reduced in treated tissue samples (Fig. 6g, h). As PXS-6302 reduces lysyl oxidase activity, and thereby the crosslinking of collagen, this results in more soluble, easily degradable extracellular matrix. As bleomycin also stimulates the expression of degrading enzymes such as matrix metalloproteases, the significant drop in hydroxyproline and immature crosslinks could

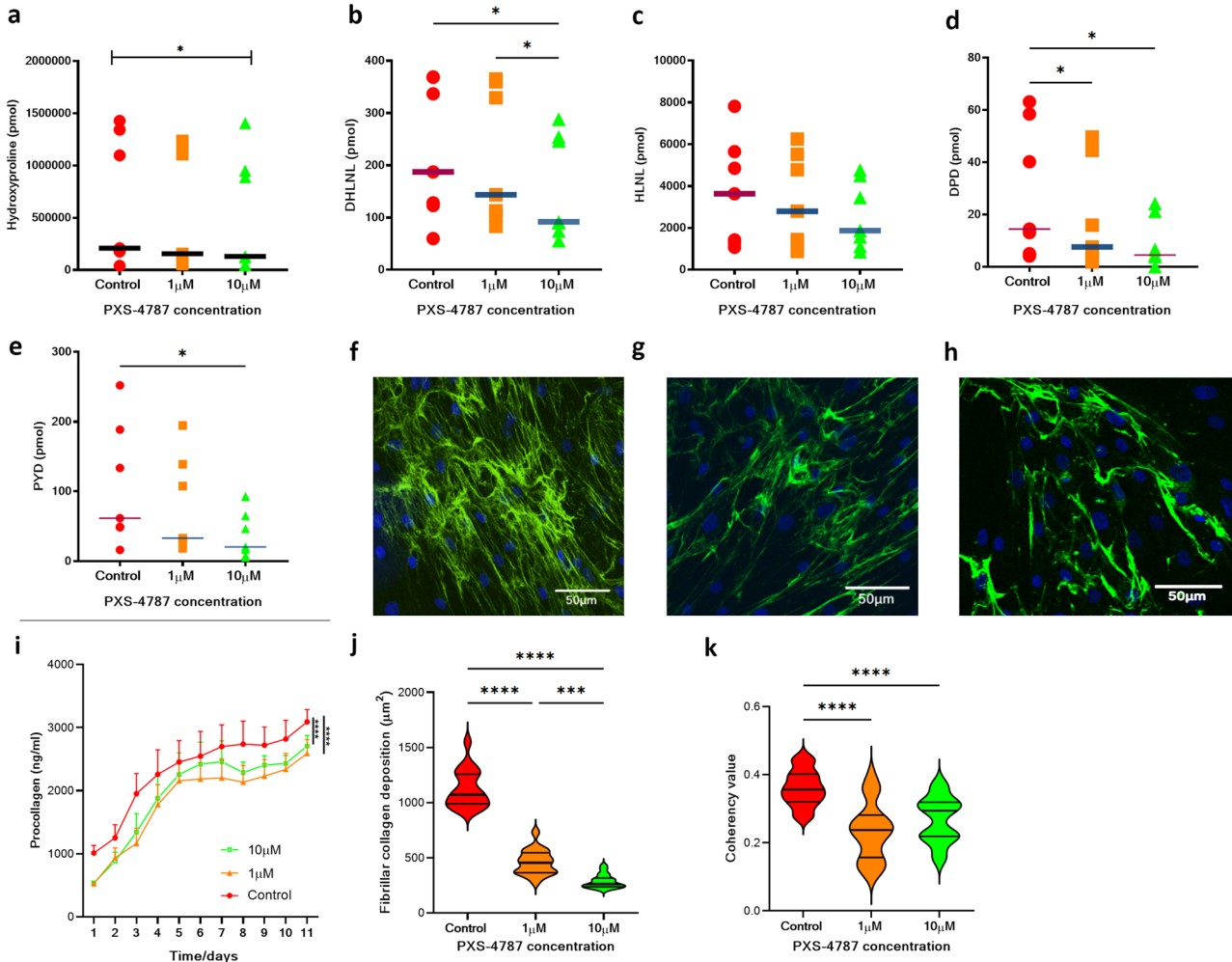

**Fig. 4 | PXS-4787 treatment reduces collagen formation, deposition and crosslinking in vitro.** Reduced collagen crosslinks formation in primary human dermal fibroblasts cultured in vitro after treatment with PXS-4787 at 0, 1, and 10 μM concentrations under scar-in-a-jar conditions for 11 days, $n = 6$ biological replicates. Reduction in (**a**) Hydroxyproline ($p = 0.0385$), (**b**) DHLNL ($p = 0.0159$, $p = 0.0165$ for control vs 10 μM and 1 μM vs 10 μM, respectively), (**c**) HLNL, (**d**) DPD ($p = 0.0359$, $p = 0.0396$ for control vs 1 μM and control vs 10 μM, respectively), and (**e**) PYD ($p = 0.0403$ for control vs 10 μM). Using a 'scar-in-a-jar' model, collagen I expression is significantly decreased ($p < 0.001$, **f–j**) when measured using a soluble procollagen I fragment detection fluorescence measure (**i** ($p < 0.0001$)) and

immunohistochemical staining (**j** ($p < 0.0001$)). Coherency of collagen I is also significantly lower (**k**) after treatment with PXS-4787 at 0, 1, and 10 μM, ($n = 6$ biological replicates). **f–h** Representative immunofluorescence staining images of collagen I staining of human dermal fibroblasts after treatment with PXS-4787 at (**f**) 0, (**g**) 1, and (**h**) 10 μM concentrations [Anti-COL1A1 antibody (Green) and DAPI (blue)], $n = 3$ biological replicates with minimum two technical replicates each sample, scale bar 50 μm. Subsequent statistical analysis was performed with one-way ANOVA with Tukey's method for multiple comparisons. *$p < 0.05$, **$p < 0.01$, ***$p < 0.001$, ****$p < 0.0001$. Source data are provided as a Source Data file.

be related to both the reduced production as well as the accelerated degradation of these matrix proteins.

## Topical application of PXS-6302 reduces collagen deposition and crosslinking in a murine model of injury

Having established promising in vivo anti-fibrotic efficacy, we next evaluated a full-thickness excision injury model in mice. Full-thickness excision injuries of approximately 1.1 cm² (approximately 2% body surface area) of dorsal skin were created under anesthesia. From Day 2 post-injury, animals received daily application of 0, 0.5, or 1.5% PXS-6302 oil in water cream for 28 days. At the end of the experiment, all animals were euthanised and scar tissue analyzed for collagen and cross-link content. PXS-6302 application showed a dose-dependent reduction of the hydroxyproline concentration, suggestive of a reduction in collagen deposition (Fig. 7a). Mature and immature (Fig. 7b–e) crosslinks were also dose-dependently diminished. Therefore, when applied after injury and during the healing phase, PXS-6302 reduced collagen crosslinking and total collagen deposition in the scar.

Reassuringly, the efficacy of PXS-6302 was in line with earlier studies (performed with the first-generation inhibitor, PXS-4787, Supplementary Fig. S7) albeit with what appears to be a more comprehensive impact on crosslinking at a lower dose.

## Topical application of PXS-6302 improves scar appearance with no reduction in tissue strength in porcine models of excision and burn injury

To confirm the anti-scarring effects of PXS-6302 in a skin model that more closely resembles human physiology, a porcine excision injury model was used. Eight full-thickness excisions (10 cm²) were made on 5 female juvenile pigs (each weighing 18–20 kg) for a total of 40 injury sites. Following on from an earlier experiment (performed with PXS-4787, Supplementary Fig. S8, S9, S10) that examined the impact of timing of inhibitor application, we next examined the effect of dose. Specifically, doses of 0, 0.5, 1.5, and 3% PXS-6302 oil in water cream were applied to the excision injury site once daily, starting 2 weeks post-surgery and continuing for 12 weeks (Supplementary Fig. S9, S10).

LOX activity was significantly reduced when measured in tissue samples isolated 24 h after the final dose (at time of euthanasia, Fig. 8a). Photos of all scars were assessed by plastic surgeons blinded to the treatment group. Independent scoring of each scar (where each scar was scored on a scale of 1–10, poor to excellent) showed significantly higher scores for the 3% treated scars suggesting significant improvement in scar appearance (Fig. 8b-d, S10).

Given the significant and positive effects observed with 3% PXS-6302 cream on the appearance of scars following excisional injury, we also tested this concentration in a porcine model of burn injury. Briefly, four deep-dermal contact burn wounds of approximately 50 cm² area

were created on the back of a 20–25 kg pig. On Day 3 post-injury, wounds were debrided by a burn surgeon and dressings replaced. Either placebo or 3% PXS-6302 cream was applied to the wound once daily from the time of re-epithelialisation until 24 h prior to euthanasia (2 months post-injury). Again assessment of scar photos by plastic surgeons blinded to the treatment group scored treated scars significantly better than their placebo-treated controls (Fig. 8e, Supplementary Fig. S10). At euthanasia, scars were excised and tensile strength and Young's modulus (YM) calculated using a Shimadzu tensile tester to investigate impacts on pliability and tissue strength of the use of PXS-6302. All scars were tested in the anterior-posterior alignment. Stress–strain curves were generated (Fig. 8f). Young's modulus showed a strong trend of reduction in treated scars, with 5/6 treated scars recording a lower YM than their matched vehicle treated scar (Fig. 8g, $p = 0.15$), suggesting PXS-6302 treated scars are more pliable than their placebo-treated controls. Tensile strength showed no significant differences between treated and control scars (Fig. 8f–h), suggesting inhibition of LOX enzymes does not lead to the fragility of scar tissue.

## Discussion

Scar formation associated with wound healing involves increased collagen deposition and crosslinking, leading to abnormal reorganization of the ECM. Aberrant and excessive deposition of type I collagen in the ECM is key to the development and maintenance of hypertrophic scars[23]. Underpinning this is the relative stability of mature cross-linked collagen and its resistance to degradation. Therefore, limiting the extent of crosslinking during scar formation may enhance scar appearance and pliability.

The clear clinical need for therapeutics to ameliorate scarring has driven drug development approaches. Activation of the TGFβ3 pathway (avotermin, discontinued 2011) and inhibition of CTGF signaling[24] did not provide significant clinical benefits. It is possible that targeting fibrosis in the active wound healing phase, and upstream of the ECM generation, may be less efficacious due to the extensive redundancies in skin repair mechanisms. Targeting of collagen crosslinking, as the final step in stable collagen deposition in the ECM, has the potential to be more successful. Alternative approaches, including microRNA-29,

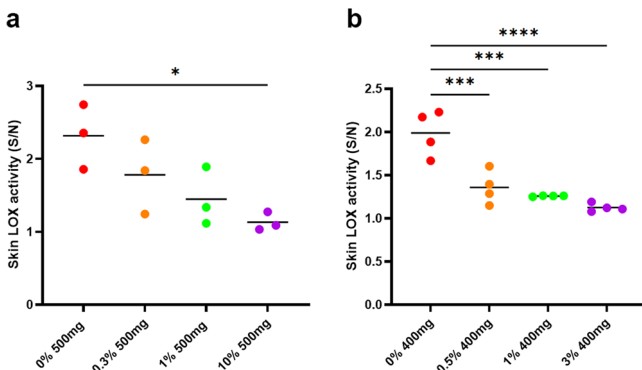

**Fig. 5 | Pharmacokinetic-pharmacodynamic measurement of target engagement in rat skin in vivo.** Dose-dependent reduction of lysyl oxidase activity in rat skin following a single dose (**a**) ($p = 0.0269$) or (**b**) repeated application of cream ($p = 0.0007$, $p = 0.0002$, $p < 0.0001$ for 0% vs 0.5%, 1% and 3% respectively). Activity was measured 24 h after last topical application. Noise in these assays was determined by measuring background fluorescent changes in the presence of a high (>300 μM) concentration of BAPN to block all lysyl oxidase activity in the presence of other amine oxidase inhibitors ($n = 3$ per group). The signal (activity in the absence of BAPN) over noise (signal of the same extract in the presence of BAPN). Statistical analysis was performed with one-way ANOVA with Tukey's method for multiple comparisons. *$p < 0.05$, **$p < 0.01$, ***$p < 0.001$ and ****$p < 0.0001$. Source data are provided as a Source Data file.

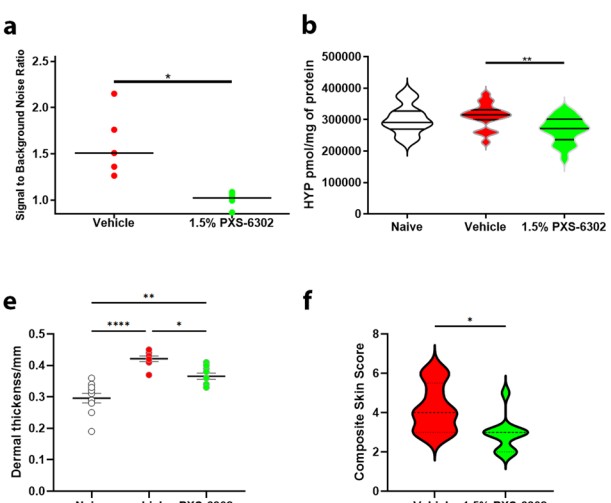

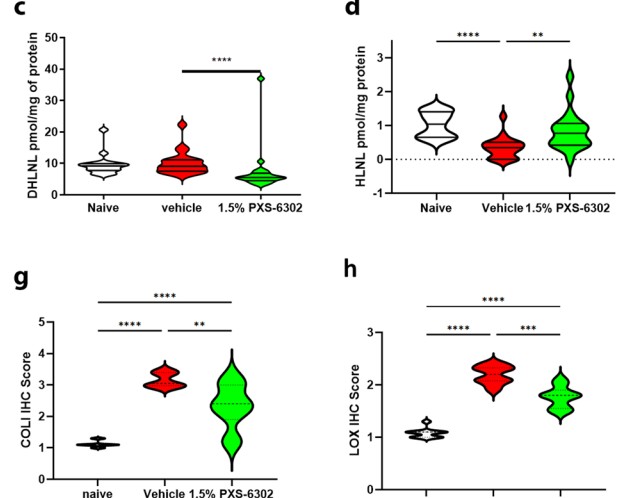

**Fig. 6 | Topical PXS-6302 is effective in reducing bleomycin-induced skin fibrosis.** Topical treatment with 1.5% PXS-6302 cream significantly inhibited LOX activity (**a** ($p = 0.001$)), reduced hydroxyproline (**b** ($p = 0.048$)) and immature DHLNL (**c** ($p < 0.0001$)) and HLNL (**d** ($p < 0.0001$)) crosslinks in the skin and reduced fibrosis as assessed by dermal thickness (**e** ($p = 0.0139$ and 0.0013 vehicle and naïve vs PXS-6302 treated respectively)) and Composite skin scores (**f** ($p = 0.0114$)). Immunohistochemistry also showed reduced Col I (**g** ($p = 0.0006$))

and LOX (**h** ($p < 0.0001$)) positive staining in 1.5% PXS-6302 treated tissue sections. Naïve animals were not exposed to bleomycin, whilst vehicle and PXS-6302 groups were treated with bleomycin to induce fibrosis ($n = 9$–10 per group). Statistical analysis was performed with two-tailed Mann–Whitney test or one-way ANOVA with Tukey's method for multiple comparisons. $p$ values are <0.05 (*), <0.01 (**), <0.001 (***) and <0.0001 (****). Source data are provided as a Source Data file.

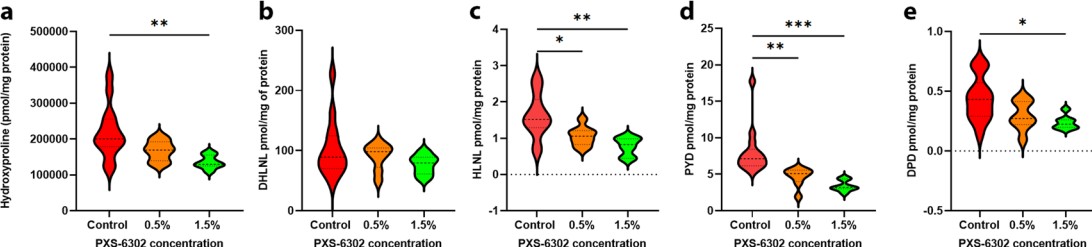

**Fig. 7 | Topical treatment with PXS-6302 reduces collagen deposition and crosslinking in a murine model of excision injury.** Hydroxyproline is significantly reduced in 1.5% PXS-6302 treated scar tissue (**a** ($p = 0.0073$)). Reduced immature crosslinks in treated groups, with a trend for (**b**) decreased DHLNL and (**c**) significantly reduced HLNL links ($p = 0.0284$, 0.0016 control vs 0.5% and 1.5%, respectively). Mature crosslinks are significantly reduced in the treated scar tissue with both (**d**) PYD ($p = 0.0081$, $p = 0.0006$ c vs 0.5% and 1.5%, respectively) and (**e**) DPD links reduced ($p = 0.0168$ ($n = 7$–14 per group)). Statistical analysis was performed with one-way ANOVA with Tukey's method for multiple comparisons, *$p < 0.05$, **$p < 0.01$, ***$p < 0.001$, ****$p < 0.0001$. Source data are provided as a Source Data file.

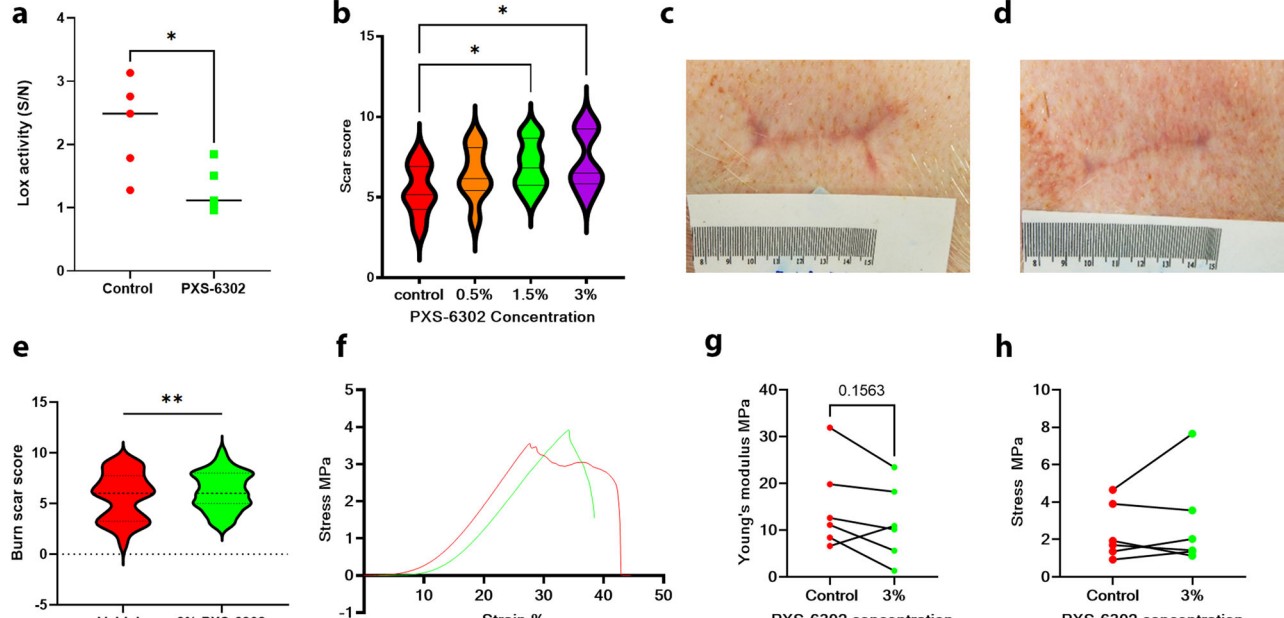

**Fig. 8 | Topical treatment with PXS-6302 inhibits LOX, reduces crosslinking and improves scar appearance in porcine models of excisional and burn injury.** LOX activity is significantly inhibited in porcine scar 24 h after final daily PXS-6302 application (total duration of application once daily for 10 weeks (**a** ($p = 0.0292$)). **b** Independently scored scars on a scale of 1–10 (poor to good scar) by plastic surgeons blinded to treatment show significantly higher scores for the 3% PXS-6302 treated scars in excision injury model ($p = 0.0028$, 0.0334 3% treatment vs control and 0.5% treatment respectively). Images of vehicle (**c**) and 3% PXS-6302 (**d**) treated scar after excision (**e**) Scars after burn injury were also significantly improved compared to placebo-treated control when scored independently by plastic surgeons blinded to the treatment group. (**f** ($p = 0.008$)) Stress–strain curves of treated (green) and control (red) scars were similar, with a reduction in Young's modulus in 5/6 treated scars compared to matched controls (**g**, **h**) and no difference in tensile strength between the two groups ($n = 10$ excision wounds each treatment group, $n = 12$ burn injuries in treated and control groups). Statistical analysis was performed with paired Mann–Whitney test or repeated measures ANOVA with Tukey's method for multiple comparisons, *$p < 0.05$. Source data are provided as a Source Data file.

are now currently being trialed for scarring (NCT03601052). However, cutaneous injections such as those required for the delivery of therapeutic in the microRNA-29 study pose significant challenges for effective delivery, especially in the context of larger wounds. These can be overcome with the use of a highly permeable anti-fibrotic cream formulation as demonstrated with PXS-6302.

Lysyl oxidases are responsible for the oxidation of lysine side chain residues that spontaneously react to form crosslinks. These crosslinks render collagen and elastin less susceptible to degradation and this increased ECM stability promotes fibrosis. Our study shows that lysyl oxidase enzymes are expressed in scar tissue, supporting the development of a pan-LOX inhibitor to ameliorate scarring. Moreover, the development of a topically deliverable lysyl oxidase inhibitor in an easily applicable cream formulation overcomes the poor compliance

of invasive therapies and the associated need for multiple hospital visits.

Here we describe the small molecule pan-LOX inhibitors, PXS-4787 and PXS-6302, that fulfill all the criteria of mechanism-based inhibitors and belong to a novel chemical class. They combine the fluoroallylamine portion of known amine oxidase inhibitors[16,25] with a phenylsulfone group to achieve selectivity and specificity. Both compounds irreversibly inhibit all lysyl oxidases and, subsequently, collagen crosslinking and deposition. The effect of both compounds is very specific, with no apparent off-target activity. This differentiates them from the typically used pan-LOX inhibitor BAPN which is a substrate for other amine oxidases, resulting in the production of reactive oxygen species in different tissues, including the vasculature.

The in vitro and in vivo effects of these inhibitors have been well characterized. In vitro, a change in the ratio of soluble to insoluble collagen occurs[26]. In fibroblast cell culture, moderate inhibition mediated by PXS-4787 did not affect the total collagen concentration but significantly reduced the concentration of immature and mature crosslinks. This reduction in crosslinks may reduce collagen fibril strength[27] and tissue stiffness[28], suggesting that in vivo this approach may lead to reduced mechanical stress that in turn will positively impact on fibroblast phenotype to sustain reduced collagen synthesis[29].

In vivo, topical once-a-day application of lysyl oxidase inhibitors [both first (PXS-4787) and second (PXS-6302) generation)] improved the appearance of scar in relevant in vivo models, indicative of a target-driven, as opposed to compound-specific, effect. PXS-6302 is ideally suited for topical application and therefore represents a pharmacological treatment that may be used either as an alternative or adjunct to current approaches, including steroid injections, laser therapy, massage, intense pulsed light and the use of compression garments[30]. We have investigated the impact of timing of the drug application in the pig excision model (using PXS-4787), with results suggesting that intervention late (3 weeks) after injury was similarly if not more effective than earlier administration. This provides a wide therapeutic window, in contrast to many approaches targeting growth factors or other stimuli critical in the acute phase of repair. This is likely to be particularly important in the treatment of larger wounds, providing an opportunity for acute intervention and re-epithelialisation prior to treatment to ameliorate scarring. In both pig (excisional and burn injury) studies we consistently found a significantly improved scar appearance upon drug administration for 3 months. Blinded scoring of wounds is an excellent method as it can be applied in pre-clinical and clinical studies with a very good translation from one to the other, suggesting these findings have a high likelihood of being reproduced in clinical trials[31]. Importantly, tensile strength of scar tissue did not appear to be impacted by the application of PXS-6302, whilst pliability trended towards improvement. Appearance and pliability of scar are the most important factors for patients after significant injury, and the potential to improve both with the use of a topical cream will provide extensive benefits for both patients and health service delivery for scar revision.

Whilst these studies suggest strong clinical potential for PXS-6302 to ameliorate scarring, it is prudent to consider the potential limitations. Animal models of scarring generally heal better than humans, and the injuries were left to heal without current best practice clinical interventions (that may reduce the impact of this compound in human trials). In addition, whilst the rodent model of fibrosis demonstrated positive results, the limitations of these models are well-known, meaning the potential impact of pan-LOX inhibition on progressive fibrosis remains unclear. We have also not measured the impact on elastin and elastin crosslinks, which may contribute significantly to changes in scar tissue. A phase I clinical single dose-escalation and repeated fixed dose trial has been completed demonstrating strong target engagement and long-lasting inhibition (ACTRN1262100322831). SOLARIA2, a Phase 1c, 3-month study in patients with established scars is currently ongoing (ACTRN12621001545853). This trial will be critical to establish the safety of PXS-6302 in humans and has the potential to show signs of efficacy. On successful clinical development the promise of PXS-6302 is to provide a new paradigm in the treatment of scars, based on the critical importance of lysyl oxidases to excessive ECM formation and maintenance, the ability to self-administer treatment and the efficacy when applied long after injury.

## Methods

### Fluorometric enzymatic activity assays
The measurement of the enzymatic activity of all lysyl oxidase family members was based on the detection of hydrogen peroxide with an Amplex-Red oxidation assay, as previously described[32]; miniaturized in 384 well formats, with the appropriate combination of substrate and assay buffer for each individual enzyme. After a 30 min pre-incubation of the enzyme at 37 °C with the test compound (or with different incubation times for the time-dependency assay), one volume of reaction mixture containing 120 μM Amplex Red (AR) (Life Technologies), 1.5 U/mL horseradish peroxidase (HRP) (Sigma-Aldrich) and the specific substrate was added to each sample. The relative fluorescence units (RFU) were then measured every 2.5 min for 30 min at 37 °C, excitation 565 nm and emission 590, on a BMG Clariostar Microplate Reader. All the substrates were used at concentrations corresponding to their Km towards the corresponding enzyme, or with a range of different concentrations for the substrate competition experiments.

Lysyl oxidases assay buffer contained 1.2 M urea, 50 mM sodium borate buffer, pH 8.2 and 100 μM β-aminoproprionitrile (BAPN, Sigma-Aldrich) was used for low control.

Recombinant human LOXL1 was expressed and purified from cDNA (GeneArt) and 10 mM putrescine was used as substrate. Recombinant human and mouse LOXL2 (R&D Systems) were challenged with 5 mM putrescine (Sigma-Aldrich) as the substrate. Rat and dog recombinant LOXL2 were amplified and purified in house from HEK-293 clones kindly provided by Dr. Fernando Rodríguez Pascual, Madrid University, and used in the same conditions as for the human form. 2 mM putrescine was used as the substrate for recombinant human LOXL3 (R&D Systems) and LOXL4 (Dr. Fernando Rodríguez Pascual, university of Madrid, Spain) assays.

### Jump dilution assays
The measurement of the residence time was based on the detection of hydrogen peroxide with an Amplex-Red oxidation assay, as previously described[33] in a 96 well format. The target was incubated with the test compound at 30 × IC50 for 40 min at 37 °C in snapstrip PCR vials. After the incubation, a 100-fold dilution is carried out in lysyl oxidase assay buffer into the vials. The diluted enzyme-inhibitor complex is added to a 96-well plate followed by the addition of the reaction mixture (20 mM putrescine for LOXL1, 10 mM putrescine for LOXL2). The target percentage activity is measured as a function of time after the dilution event.

### Compound oxidation assay
This assay determines the substrate propensity of a compound relative to background (dimethyl sulfoxide only). Compound oxidation by rhSSAO was measured by fluorometric assay[34]. Briefly, rhSSAO was incubated for 2 h at 37 °C in HEPES buffer before the addition of an equal volume of Amplex Red (20 mM), horseradish peroxidase (4 U/ml), and test compound in the same buffer. The kinetics of the formation of resorufin was measured immediately using Optima reader (BMG Labtech GmbH, Ortenburg, Germany), at 37 °C.

### Tissue immunohistochemistry for LOX
Formalin fixed paraffin embedded (FFPE) skin tissue sectioned were cut to 4 μm sections using a Leica RM2235 microtome. Sections were placed on a Superfrost plus slide and allowed to incubate for 2 h at 60 °C. Sections were dewaxed and heat-induced antigen retrieval conducted. Slides were stained using the Bond Dewax solution kit (Leica AR9222) with Rabbit anti-human LOX antibody (Abcam, AB31238) diluted 1:2000 with Bond Primary Antibody Diluent (Leica AR9352). Sections were then processed using the Leica BOND polymer refine detection kit (DS9800) After immunohistochemistry protocol was complete slides were counterstained with Haematoxylin and cover-slipped.

### Gene expression using qPCR
Comparison of LOX, LOXL1, LOXL2, LOXL3, LOXL4 in scar and normal skin fibroblasts used cells isolated from matched patients (n = 15) and

seeded at a density of 250,000 cells/ well in a 6-well plate and incubated for 16 h. After 16 h media was changed and cells incubated for 48 h. RNA was extracted using RNeasy mini kit according to the manufacturer's protocol (qiagen GMbH) and concentration measured on a nanodrop. The cDNA was synthesized using quantitect reverse transcriptase kit from qiagen. 100 ng of cDNA was analyzed using Quantitect sybr green (from Qiagen) using LOX (Cat no. QT 00017311), LOXL1 (Cat no00011830, Qiagen), LOXL2 (Cat no. QT 00019425, Qiagen), LOXL3 (Cat no. QT 00066556, Qiagen), LOXL4 (Cat no. QT 00046466, Qiagen) quantitect primer. Two housekeeping genes b-actin and GAPDH were used to normalize collagen 1 and LOX family gene expression using GAPDH quantitect primers (Cat no. QT 00079247, Qiagen) and β-actin quantitect primers (Cat no. QT 01680476, Qiagen). The data were analyzed using the Rotor geneQ software and the relative gene expression was calculated using the $2^{-(\Delta\Delta c\ (t))}$ method[35]. Three biological samples were used for each condition.

Changes in COL1A1 and LOX gene expression in PXS-4787 treated cells were done using primary human dermal fibroblasts (50,000) seeded in a 6-well plate under Scar-in-a-jar media. The cells were incubated with 0 μM, 1 μM, and 10 μM PXS-4787 treatment for 48 h. mRNA was extracted using RNeasy mini kit® according to the manufacturer's protocol (Qiagen Gmbh). The total concentration and purity of RNA were determined using a NAnodrop 2000C (Thermo Scientific, USA). For reverse transcription, mRNA was converted to cDNA using QuantiTect Reverse Transcription kit (Qiagen; Catalog no. 205311) according to the manufacturer's protocol. Quantitative real-time RT-PCR was conducted using SYBR Green Master Mix (Qiagen; Catalog no. 204143) and Rotor gene Q thermocycler (Qiagen). The reactions were carried out in triplicate in 20 μl total volume containing 1 μl cDNA and 2 μl primers. The QuantiTect Primer assay for COL1A1 (Catalog no. QT00037793), LOX (Catalog no. QT00017311) were used. GAPDH (Catalog no. QT00079247) and PGK1 (Catalog no. 249900) were used as a housekeeping gene. The data were analyzed using Rotor geneQ software and the relative gene expression was calculated using the $2^{-(\Delta\Delta c\ (t))}$ method[35]. Three biological samples were used for each condition.

## Cell culture

Human primary dermal fibroblasts derived from human skin were cultured in Dulbecco's Modified Eagle's Medium (DMEM-Glutamax™; Invitrogen Gibco) supplemented with 10% fetal bovine serum (FBS; Invitrogen Gibco) and 1% Penicillin/Streptomycin (Invitrogen Gibco). The cell culture maintained in a humidified chamber at 37 °C and 5% $CO_2$. All primary cells were obtained from discarded skin after elective surgery with informed consent. Collection was approved by South Metropolitan Health Service Ethics committee (RGS00000099).

## Cell viability assay

Cell proliferation was measured using a standard 3-(4,5-dimethylthiazol-2-yl)-5-(3-carboxymethoxyphenyl)-2-(4-sulfophenyl)-2H-tetrazolium, inner salt (MTS) assay (Cell Titer 96® Aqueous One Solution Cell Proliferation Assay, Promega, Australia) as per the manufacturer's protocol. Briefly, 2500 primary scar fibroblasts below passage eight were seeded in each well of 96-well plates. The cells then exposed to 0, 1, 2.5, 5, 10, 25, 50, and 100 μM concentrations of PXS-4787 in cell culture media (DMEM-Glutamax™ supplemented with 10% Fetal Bovine Serum and 1% Penicillin/ Streptomycin, Life technologies, USA) for 24, 48 and 72 h. MTS solution (20 μL) was added in each well and incubated for 3.5 h in a humidified chamber at 37 °C and 5% $CO_2$. The absorbance recorded on a UV−visible spectrophotometer (Enspire Multimode plate reader, PerkinElmer®) at 490 nanometers.

## Scar-in-a-jar

An experimental 'Scar-in-a-jar' protocol developed by Chen and Raghunath[36] was followed for quantitation of collagen per cell and to measure the orientation of collagen under fibrotic conditions in vitro. Primary scar fibroblasts below passage eight were seeded on four-well chamber slides (Nunc™ Lab-Tek® Thermofisher Scientific, USA) with seeding density $5 \times 10^4$ cells per well and cultured in normal media (DMEM-Glutamax™; Invitrogen Gibco supplemented with 10% FBS and 1% penicillin/streptomycin). The cultured cells were allowed to incubate in a humidified chamber at 37 °C and 5% $CO_2$ for 16 h, after which the media was changed to a stimulating media mixture, containing crowding molecules: 375 mg/mL Ficoll 70 and (25 mg/mL) Ficoll 400 (Sigma Aldrich, USA), 5 ng/mL TGF-β1 (R & D Systems, USA), 0.5% FBS, 100 mM L-ascorbic acid-2-phosphate (Sigma Aldrich, USA) in DMEM-Glutamax™ supplemented with 0.5% FBS and 1% Penicillin/ Streptomycin (Life Technologies, USA) and 1 and 10 μM concentrations of PXS-4787 for 6 days. After 6 days, cells were blocked with 3% bovine serum albumin in Fluorobrite® DMEM. Primary antibody solution Collagen 1 (1:1000 in 3% BSA in Fluorobrite®, mouse monoclonal IgG1, Cat. No. C-2456, Sigma-Aldrich, USA) was added and incubated at 37 °C and 5% $CO_2$ in a humidified chamber for 90 min followed by washing three times with Fluorobrite®. The cells were fixed with 4% paraformaldehyde solution for 10 min at room temperature followed by three washes with Fluorobrite®. The secondary antibody (goat-anti mouse Alexa-fluor 488 (1:500 dilution), Cat. No. A-11001, Life Technologies, USA) Hoechst® staining solution (Cat. No. H3570, Life Technologies) were incubated for 30 min and cells were washed with Fluorobrite. The coverslips were finally mounted on the slides using Prolong® Gold anti-fade mounting solution (Life Technologies, USA). The slides were imaged on the confocal microscope (Nikon A1Si Confocal Microscope,40x oil immersion objective, 1.3 aperture, laser 2). Three separate 3D images were collected from each slide at 488 nm laser wavelength using 0.05 μm interval with a resolution of 1024 ×1024 pixels and then condensed into a single image using the maximum projection function. The images then further analyzed for collagen coherency measurements using 'Fiji is just image j's' (FIJI) orientation J application as previously reported[18,37]. For quantitation of collagen per cell, an image of whole chamber slide was captured using fluorescence microscope (Nikon inverted microscope). The 488 nm green and Hoechst stained blue images were collected, and analyzed using NIS-Elements software (Nikon, Japan). The six ROIs were selected from each image and the binary thresholds were recorded for each corresponding image. The threshold of the collagen fibers (green image) was divided by the threshold of the cell nuclei (blue image) to calculate the amount of collagen per cell.

## Procollagen measurements

Fluorescence Resonance Energy Transfer (FRET) assay was conducted using a Human Pro-Collagen Type 1 Kit (Cisbio Bioassays, Codolet, France; Cat. No.63ADK014PEH) and following the manufacturers' protocol ($n = 7$). The primary dermal fibroblasts were set up for the Scar-in-a-jar assay as described above and maintained in the culture for 11 days. The supernatant (50 μL) was collected every day. Each sample was then diluted at a 1:5 ratio in DMEM media, 16 μL of the sample was added to 2 μL of Anti-Human procollagen acceptor antibody and 2 μL of Anti-Human procollagen donor antibody, followed by incubation for 3 h at room temperature. The plate was read on a BMG Clariostar plate reader at the wavelengths of 620 nm and 665 nm. The ratio of acceptor and donor emission signals, the percentage coefficient of variation (%CV) and the delta F (%) were all calculated as per kit manufacturer's instructions. A standard curve was generated and the experimental values were calculated from the standard curve.

## Collagen crosslinking in vitro

Primary scar fibroblasts were seeded in a complete media at density $75 \times 104$ cells per well in a T-25 tissue culture flask at 37 °C and 5% $CO_2$. After 16 h the complete media was replaced by the stimulated media (375 mg/mL Ficoll 70 and (25 mg/ mL) Ficoll 400, 5 ng/mL TGF-β1, 0.5% FBS, 100 mM L-ascorbic acid-2-phosphate) in DMEM-Glutamax™ supplemented with 0.5% FBS and 1% Penicillin/ Streptomycin and 1 and 10 µM concentrations of PXS-4787 for 11 days to allow time for sufficient crosslinking of extracellular collagen. After 11 days, the media was aspirated and cells were trypsinized for 30 min and reduced with 12 mg/mL Sodium Borohydride in 20 mM Sodium Hydroxide solution followed and incubated for 30 min at room temperature. The pH of the solution was made acidic using 50 µL of neat acetic acid. The solution was centrifuged at 20,000g for 15 min at 4 °C followed by three washes with MilliQ water and samples processed for crosslinking analysis.

## Collagen crosslinking analysis for in vitro and in vivo tissue samples

Cell pellets or isolated tissue samples were completely hydrolyzed in 1.5 mL 6 M hydrochloric acid for 24 h at 105 °C. After complete hydrolysis, samples were dried using a vacuum dryer and reconstituted using 200 µL MilliQ water. The sample clean-up was performed using solid-phase extraction method which uses C18 and SCX solid-phase extraction cartridges for sample purification. The samples were run on UPLC-MS-MS (Mass Spectrometer: Thermo Scientific TSQ Endura Triple quadrupole, UPLC: Thermo Scientific Dionex UltiMate 3000) using Agilent RRHD SB-C18,2.1x50mm,1.8um analytical column and solvent A: 10 mM ammonium formate, 0.1% formic acid, 0.1% HFBA in $H_2O$ and solvent B: 10 mM ammonium formate, 0.1% formic acid, 0.1% HFBA in MeOH and analyzed as previously described[38]. Limits of detection are listed in Supplementary Table S4.

## Transcriptome analysis

Normal skin samples from consenting patients undergoing elective plastic surgery were previously collected from the South Metropolitan Health Services (ethics approval number: RGS000000099). Six different normal skin patient samples were used. To allow for transcriptome analysis in samples of normal skin fibroblasts and keratinocytes samples ($n = 7$), RNA extraction was conducted using the RNeasy Mini kit (cat. no. 74104; Qiagen, Netherlands). Fibroblasts and keratinocytes of low passage (p2-5) were cultured onto T25 flasks until ~70% confluence was reached. One flask of each fibroblasts cell line was treated with 10 µM PXS-4787 and another with complete fibroblast growth media for 48 h at 37 °C. One flask of each keratinocyte cell line was treated with 10 µM PXS-4787 and another with basal keratinocyte medium supplemented with 1.2 mM calcium chloride (Sigma; c7902) for 48 h at 37°. Cells were then collected using 0.05% trypsin and washed in Phosphate Buffered Saline (PBS), before RNA was extracted. RNA was extracted using the RNeasy Mini kit protocol and stored at -80 °C for further analysis. A NanoDrop™ 1000 Spectrophotometer (Thermo Fisher Scientific, USA) was used to examine the quality and quantity of the extracted RNA. Quality control tests of the samples were also conducted using the Agilent Bioanalyser and LabChip GX software (v4.2.1745.0) at Harry Perkins. Samples were then sent to the Australian Genome Research Facility (AGRF), where 1µg of RNA was submitted for next-generation sequencing. Illumina NovaSeq Control Software (NCS) (v1.6.0) for image analysis and Real Time Analysis (RTA) (v3.4.4) for real-time base calling was used, producing a 100 bp single end run. The Illumina bcl2fastq 2.20.0.422 pipeline was used to generate the primary sequence data. Once returned, FastQC (v0.11.3) was used to check the quality of the RNA sequencing raw data and reads of poor quality (the first 15 bp) were trimmed using the fastp software[39]. The RNA sequencing raw reads were then aligned using the STAR read aligner (v2.7)[40] to the Ensembl release 98 Human GRCh38.p13 reference genome (http://asia.ensembl.org/Homo_sapiens/Info/Index)[41]. SAMtools (v1.9)[42] was used to generate BAM files. Exploratory analysis (sample-to-sample distribution and prinPCA) used rlog transformed values. Differential gene expression analysis was run using DESeq2 package (v1.24.0)[43] on R (v3.6.1) with p.adj <0.01 cut-off to determine significantly differentially expressed genes.

## Lysyl oxidase activity inhibition assay

Human skin samples were maintained in DMEM Glutamax™ supplemented with 10% FBS and 1% Penicillin/Streptomycin and amphotericin/kanamycin until treatment. Two 5 mm² pieces of skin tissue were treated with vehicle or vehicle plus drug for 4 h at 37 °C and the measurement of the enzymatic activity performed as previously described[44]. Briefly, after freezing in liquid nitrogen, the skin was sliced using a cryostat (Leica CM3050) to sections of 100 µm thickness and incubated at room temperature in PBS containing protease inhibitors (Protease inhibitor cocktail I, 539131, Merck) for 2 h. For the assay, 100 µL of 1.2 M urea sodium borate buffer was transferred into 24 wells of a 96-well plate. Slices were transferred into wells containing 100 µL of 1.2 M urea sodium borate buffer and the plate was incubated at 37 °C for 15 min. For each assay well, 100 µL of the reaction mixture containing 120 µM Amplex red, 1.5 U/mL Horseradish Peroxidase and 20 mM Putrescine. Next, 4 µL 100 µM PXS-4787 and 100 µM BAPN was added into the respective wells. The relative fluorescence value was measured on a microplate reader every 2.5 min for 30 min at 37 °C, excitation 565 nm and emission 590 nm and data expressed as the ratio of signal in the absence of BAPN and signal in the presence of BAPN.

## Franz Cell skin permeability assay

Drug diffusion studies were performed using tailor-made Franz™ diffusion cells (JR and SR Davis, Australia). Briefly, human skin obtained from elective plastic surgery procedures (ethics approval by South Metropolitan Health Service RGS00000099) was washed in PBS and any subcutaneous fat removed using a scalpel. Pieces of full-thickness skin were mounted between the donor and receptor compartments of the Franz cell with the epidermal layer facing the donor chamber and the dermis immersed in PBS in the reservoir (receptor chamber). After fifteen minutes pre-incubation at 30°C in an air incubator sufficient cream (vehicle) containing 3% PXS-6302 (or PXS-4787) to cover the entire surface area (approx. 1cm², volume of cream applied approx. 500µl) was applied to the epidermal surface in the donor chamber. 200µl sample volumes were removed from the receiver chamber at specific intervals and the 200 ml volume replaced with PBS. Samples were then filtered through 0.22 µm filters and analyzed using LCMSMS. The concentration of PXS-6302 was determined by LC-MS/MS using a Waters Xselect CSH column, (50 mm×4.6 mm, 2.5 µm) using Isocratic HPLC with 0.1% formic acid solution in water:0.1% formic acid in acetonitrile (95:5), 0.8 ml/min flow rate with Atmospheric pressure ionization (API) mass Spectrometry.

## Pharmacodynamic rat assay

PXS-6302 was formulated as an oil in water cream of different concentrations (0, 0.3, 1, 10%) and applied to a shaved area on the back of an adult rat (500 mg cream applied to 16 cm² skin surface area). After 24 h animals were euthanased. For the detection of lysyl oxidase activity in the skin, epidermal and dermal layers of the skin were dissected, pre-cooled with liquid nitrogen and pulverized. Samples were washed by homogenization with ice-cold wash buffer (0.15 M NaCl, 50 mM sodium borate, pH 8.0 with 0.25 mM PMSF and 1 µL/mL bovine aprotinin as protease inhibitors) at 100 µL/mg, centrifuged at 10,000 × g for 10 min at 4 °C and supernatant discarded. After the final washing step, the pellet was resuspended in extraction buffer (6 M urea, 50 mM sodium borate, pH 8.2 with 0.25 mM PMSF and 1 µL/mL aprotinin as protease inhibitors) with ratio of buffer volume to tissue weight at 4:1. After a 3 h incubation at 4 °C, the mixture was diluted 1:2 with assay

buffer containing 50 mM sodium borate (pH 8.2), centrifuged at 20,000 × g for 20 min at 4 °C. The collected supernatant was spiked with pargyline hydrochloride and mofegiline hydrochloride at final concentrations of 0.5 mM and 1 μM, respectively, to inhibit amine oxidases. Lysyl oxidase activity was measured as previously described[44]. The concentration of PXS-6302 in the skin was also measured in the same samples using LCMSMS as described for Franz cell diffusion assay.

### Bleomycin-induced skin fibrosis in mice

8–10-week old C57BL/6 mice (*n* = 10 each group) were used to induce skin fibrosis. Bleomycin (Bleomycin sulfate from Streptomyces verticillicus, Euroasia) dry powder 1.5 IU/mg was reconstituted in saline at 2 IU/ml concentration. Starting on day 0 and continuing on alternate days thereafter, animals received subcutaneous injection of Bleomycin (0.1 IU in 50 μL volume) and/or saline on the right flank. For each animal, each dose was injected at a single site. The injections were continued for 20 days throughout the study. Starting from day three, vehicle cream or vehicle plus drug cream were applied topically once a daily treatment to a 2 cm × 2 cm area on the right dorsal flank of the animal for 20 days. On Day 21, all animals were euthanized and the skin tissue was collected for further histopathological analysis. The skin tissue was fixed in 4% PFA and sectioned skin was stained with Masson's Trichrome, Collagen I and LOX and used to assess dermal fibrosis. Masson's Trichrome was used to assess fibrotic index. Collagen Stain was subjectively scored for intensity as compared to control tissue. LOX Stain was subjectively scored as stain accumulation compared to control tissue. All images were assessed by assessors blinded to the treatment group. Composite skin/immunohistochemistry scoring tables used are in the Supplementary Information (Supplementary Tables S5, S6).

### Murine excision injury model

Adult C57BL6/J female mice were maintained in standard housing with food and water provided ad libitum. All experiments were approved by institutional ethics committees (UWA ethics number: R/3/100/1478) and performed as per the National Health and Medical Research Council (NHMRC) Australian Code of Practice for the Care and Use of Animals for Scientific Purposes.

Nine-week-old C57BL6/J female mice (*n* = 8 per group) received a full-thickness excision injury of 1.1 cm² on dorsal area using a circular 12-mm biopsy punch (Acu-Punch, Badand Medical, P1250) (approximately 2% body surface area) under anesthesia. All animals received intramuscular injection of buprenorphine (0.1 mg/kg) as an analgesia. Paracetamol (1 mg/ml) was provided for 5 days post-injury in drinking water. Animals were allowed to recover for 24 h post-injury. Animals in the treatment group received twice a day application of 3% PXS-4787 cream for 28 days. For PXS-6302, once per day application of 1.5% concentration cream was used. At the end of the experiment, all animals were euthanised using an intraperitoneal injection of pentobarbitone (160 mg Kg-1) under Isoflurane anesthesia, and scar tissues were removed. The scar skin was fixed in 4% paraformaldehyde overnight at 4 °C. After fixation, tissues were washed in sterile PBS at room temperature. Samples were processed in Leica ASP300S Tissue Processor followed by embedding in paraffin blocks using tissue processor (Leica EG1150 modular tissue embedding center). The paraffin blocks were sectioned at 5 μm thickness using a wax microtome (fully automated Leica RM2255). For standard histology, the skin sections were stained with hematoxylin and eosin stain and Masson's Trichrome stain. The stained sections were used to determine the collagen fiber orientation in the control and treatment specimens. All stained skin tissue sections were processed for whole slide imaging using a ScanScopeXT (Aperio Technologies, USA) at 20-x magnification. The digital images were collected and analyzed using ImageScope software (Aperio Techologies, Inc, USA). A piece of excised mouse scar tissue was processed for the collagen crosslinks extraction. The scar skin was weighed and homogenized in 900 μL PBS (FastPrep-24TM5G(MP) homogenizer). The tissue homogenate equivalent to 10 mg tissue was transferred to heat/acid resistant screw cap tube (VWR 16466-060) and the total volume of 870 μL was adjusted with PBS. The tissue homogenate was reduced with a Sodium borohydride buffer and processed for cross-link extraction protocol as mentioned in previous section.

### Porcine excision injury model

Female Juvenile pigs of 18–20 kg were housed in standard housing with food and water provided ad libitum. All experiments were approved by institutional ethics committees (UWA ethics number: R/3/100/1538) and performed in accordance with the National Health and Medical Research Council (NHMRC) Australian Code of Practice for the Care and Use of Animals for Scientific Purposes. To evaluate the effect of PXS-4787 in pigs, a surgical full-thickness excision injury model was used. All pigs (*n* = 5) from the experiment received eight full-thickness 10 cm² area excision injuries (80 cm² total body surface area, 5 × 2 cm² dimensions for each injury site). Excisions were created in one operation by a plastic surgeon whilst animals were under isoflurane anesthesia. The entire epidermal and dermal layers of skin were removed leaving excisions approximately 5–8 mm in depth. Four wounds were created along each flank creating eight injury sites per animal. Wounds were dressed and jackets provided to cover wounds (anti-anxiety jackets (Thunder shirt, RSPCA). Analgesia was given (buprenorphine injection and fentanyl patch (50 μg/hr) for 10 days post-surgery) and animals were allowed to recover. The dressings were changed at regular intervals to protect the wounds to prevent infection. In the first study, all animals received 3% PXS-4787 treatment starting 1, 2 and 3 weeks post-injury. Control wounds received the treatment from 1-week post-surgery. The treatment was continued for total 12-week period. In a second study, all animals were treated with 0.5, 1.5 and 3% concentrations of PXS-6302 cream for 12 weeks. All animals were monitored for the wound healing and scar formation throughout the study. At the end of experiment, animals were put under anesthesia and photographs were taken using a camera. All scar tissues were processed further for analysis. All photos of pig scars were shown to a minimum of 6 plastic surgeons for scoring. The surgeons ranked scars in each set of 4 (for example, a control and all three treatment groups from an individual pig and site (anterior or posterior) were compared) from best to worst (0–3 with 0 best and 3 worst in a set of 4 matched scars). The surgeons were blinded to the treatment group.

The effect of PXS-4787 topical treatment on in vivo LOX activity inhibition was measured using the method as described above. The skin sections were fixed in 4% paraformaldehyde and embedded in paraffin blocks. The 5 μm sections were stained with Masson's trichrome stain for collagen and images were collected by whole slide imaging using a ScanScopeXT (Aperio Technologies, USA) at 20-x magnification. The digital images were analyzed using the ImageScope software (Aperio Techologies, Inc, USA).

### Scanning electron microscopy

Samples of pig skin scar tissue of approximately 3 × 3 mm size, which were previously frozen in liquid nitrogen, were thawed and fixed in 4% paraformaldehyde followed by dehydration in alcohol and absolute acetone. The skin samples were further dried in a critical point dryer in liquid carbon dioxide. The specimens were then Platinum-metallized with a sputtering device. The images of the skin architecture were captured using a scanning electron microscope (Zeiss 1555 VP-FESEM, Germany). Images of collagen fibers were acquired at a 500x and 10,000x magnification. The thickness of collagen fibers was measured using ImageJ® software using the ruler and measure tool.

## Porcine burn injury model

All experiments were approved by institutional ethics committees (UWA ethics number: R/3/100/1724) and performed in accordance with the National Health and Medical Research Council (NHMRC) Australian Code of Practice for the Care and Use of Animals for Scientific Purposes. Four deep-dermal contact burn wounds of approximately 50 cm² area were created on the back of a female 20–25 kg pig (n = 3 pigs, $n$ = 12 wounds total). Burn was induced whilst under anesthesia using a brass rod of 8 cm diameter heated in a water bath for a minimum of 10 min and maintained at 100 °C. The brass rod was applied by the surgeon for 10 s duration with no additional pressure on contact. After the burn injury the wounds were dressed and pigs allowed to recover with fentanyl analgesic patches applied to a non-injured area of the skin. On Day 3 post-injury, wounds were debrided by a burn surgeon and dressings replaced. Either placebo or 3% PXS-6302 cream was applied to the wound once daily from the time of re-epithelialisation (approximately 2 weeks post-injury) until 24 h prior to euthanasia (2 months post-injury).

## Tensile strength testing

The mechanical properties of the samples were tested using a Shimadzu tensile tester (EZ-L, Shimadzu, Japan). The samples were subjected to a tensile test with a constant rate of 5 mm/min. Young's modulus was calculated from the slope of the initial part of the stress/strain curve, where the relationship between stress and strain is linear.

## Statistical analysis

All data were analyzed using Graphpad Prism software for windows. Data were presented as mean ± Standard error mean (SEM). Unpaired Student's $t$ test followed by Mann–Whitney post analysis was performed for analysis of two independent groups. Paired Students t-test followed by Wilcoxon test was used for analysis of groups with matched data. One-way ANOVA was used for comparison between more than two groups followed by Tukey's multiple comparison. Two-way ANOVA was used for comparison of more than one factors between more than two groups followed by either Sidak or Tukey's multiple comparison test. $p < 0.05$ is considered significant.

## Data availability

The data supporting the findings from this study are available within the manuscript and its supplementary information. RNASeq data is available at GSE163309. Source data file is provided with this manuscript.

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

## Acknowledgements

The authors acknowledge the facilities, and the scientific and technical assistance of Microscopy Australia at the Centre for Microscopy, Characterization & Analysis, The University of Western Australia, a facility funded by the University, State and Commonwealth Governments. The authors acknowledge the Pharmaxis staff for their support and contributions towards identifying these inhibitors. The authors also acknowledge Animal Care Services staff for their support and assistance with surgery and monitoring for both rodent and porcine studies. The work was supported by a National Health and Medical Research Council Development grant (1113736) awarded to F.M.W., W.J., K.I., and M.W.F. T.D.C. was supported by a National Health and Medical Research Council Peter Doherty—Australian Biomedical Fellowship (Grant No.: 1073180). TDC acknowledges funding from the Raine Medical Research Foundation (Raine Priming Grant) which supported this work. MWF was also supported by the Fiona Wood Foundation and the Stan Perron Charitable Foundation and the Perth Children's Hospital Foundation. Some studies were funded by Pharmaxis as part of the drug discovery efforts to identify topical anti-scarring agents. W.J., A.D.F., Y.Y., A.J. are employees of Pharmaxis.

## Author contributions

Conceptualization: M.W.F., F.M.W., W.J. Methodology: N.C., A.W.S., T.D.C., A.F., Y.Y., A.J., P.T., N.H., M.W.F., W.J., P.E.M., G.W., S.S. Investigation: N.C., A.W.S., N.H., P.T., T.D.C., S.R., P.E.M., Z.D., G.W., S.S. Funding acquisition: M.W.F., W.J., F.M.W., K.I. Project administration: M.W.F., W.J., A.D.F. Supervision: S.R., K.I., F.M.W., M.W.F. Writing: original draft: N.C., M.W.F., A.D.F., W.J. Writing: review & editing: N.C., A.W.S., T.D.C., A.D.F., Y.Y., A.J., P.T., N.H., Z.D., G.W., S.S., P.E.M., S.R., W.J., K.I., F.M.W., M.W.F.

## Competing interests

A.D.F., Y.Y., A.J., and W.J. are all employees of Pharmaxis. All other authors declare no competing interests.
