## [Peer Review File · Nature Communications]

REVIEWER COMMENTS

Reviewer #1 (Remarks to the Author):

This descriptive study discloses the identification of a new LOX inhibitor. Currently, the project is preliminary, and controls are lacking. Models are not studied in depth

1. There is no description of "iLox" in the introduction. What is the IUPAC name of this, what is it exactly, how was it discovered? How specific is it? What is its MOA? Please provide actual data. Is it a copper chelator? Was this selected out of a combichem library with BAPN as an initial structure? Does it affect non-amine oxidases?
2. Figure 3 is confusing. Please separate bleomycin and wounding data. Does inhibitor affect kinetics of wound closure in excisional punch biopsy model. Trichrome and anti- α -SMA staining would be helpful. Where is the untreated control? Please show primary data, not graphs.
3. As I understand the collagen crosslinking analysis was done on isolated fibroblasts, not tissue (methods). Please, explain Fig 3 D-H. What about crosslinking in bleomycin? Please provide a table outlining entire results of the proteomic analysis
4. Data to support effects on fibrosis are lacking eg real time PCR on key messages

Reviewer #2 (Remarks to the Author):

This is a comprehensive manuscript that presents remarkable and exciting work demonstrating unique and highly desirable features of a novel lysyl oxidase family small molecule inhibitor as a potential therapeutic agent. This has been a long-sought goal in the field. The study tests the ability of this inhibitor to attenuate skin fibrosis by topical application both in organ culture and in vivo. The basic enzymology demonstrating high specificity, efficient irreversible inactivation and lack of substrate potential against related amine oxidases is well done and convincing and well presented. Pharmacologic parameters were thoroughly evaluated. Efficacy in skin organ culture and in skin excisional wounds in vivo is also comprehensive.

A few (minor) suggestions are made to increase clarity of the manuscript. Questions should be addressed regarding whether or not elastin cross-linking was examined, and whether or not collagen fibril diameters were or could be analyzed from EM images. Parameters of MS/MS analyses and data would have been helpful to have been provided as well. Specific comments are provided below for the authors to consider.

1. Figure 2: The legend should indicate that cell layers were used for cross-link analyses. It is unclear which collagen propeptides were measured, and in media or cell layer, and what methodological approach was used. This is explained in supplementary information, but brief statements would help clarify these basic parameters in the main article.
2. "Coherency analysis" is not a very commonly used term and is not clarified in the main text. Perhaps consider a different way to describe quantitative analyses of immunofluorescence images.
3. Figure 4A control group: The legend should indicate that the time points refer to when iLOX treatment began, and the common 12 week time point of harvest of the tissues should also be indicated in the legend. Otherwise it is unclear and confusing regarding how the single control relates to the time points of the experimental samples.
4. EM images and analyses: Collagen cross-sections to measure diameters of fibrils would be of interest to determine if these are the same or different in inhibitor treated skin. Collagen diameters often increase as a function of LOX inhibitors.
5. Discussion: At the bottom of page 18 "This reduction in cross-links will reduce collagen fibril strength (32) and tissue stiffness (33)", please change "will" to "may" since no data regarding fibril strength was presented in this manuscript.
6. Methods: Crosslink analyses: Description of methods stopped at HPLC step. MS/MS conditions and data analyses strategies used should also be presented, or at least relevant respective publications cited.
7. Skin contains elastin in addition to collagen. Were desmosine and isodesmosine elastin cross-links analyzed by MS/MS, which are also dependent on LOX and can contribute to

fibrosis?

8. Reference 14 and 15 are identical.

9. Consider using PMID: 11779117 instead of reference 39.

Reviewer #3 (Remarks to the Author):

This paper describes in vitro and in vivo results of testing an anti-fibrotic agent based on inhibition of lysyl oxidases. Penetration on human skin, toxicology data and effects in reducing collagen crosslinking and reduced scarring after skin injury in different animal models are described.

The data are well described and extensive sets of experiments are presented. Some experimental details need to be added, but especially, the following points need to be addressed before this paper can be published.

1. Bleomycin model: Fig 3. Describes Bleomycin induced fibrosis. However, bleomycin is also used to reduce fibroblast proliferation eg in keloid scars. How effective was this to induce fibrosis? Treatment with the inhibitors was started already at day 3: by that time, there will not be a fully developed hypertrophic or fibrotic skin. Why was the treatment started this early and not after 20 days? Were the slides of this experiment scored blinded to treatment? Please also show histology & crosslink measures of control and bleomycin versus bleomycin iLOX treated mouse skin.

2. Full thickness wounds in mice: Please present these data in a separate figure to avoid mix-up with the bleomycin experiment.

Was wound healing delayed in alignment with lower collagen production? Was wound contraction affected? Were there effects on skin strength noted? What was the duration of the treatment? Were control wounds treated with the vehicle?

From the description it seems that the iLOX was administered on the open wounds. In that case, it would be relevant to determine the systemic load of the inhibitor, as it is likely to be higher than if penetrated through intact skin.

3. Has a validated scar assessment technique been used for the scar evaluation? Just a scoring system of better vs worse does not seem adequate; each scar should be evaluated at its own merit, not in a comparative fashion to the other treatment groups. The latter method could artificially increase differences due to the fact that raters are asked to establish a ranking where there might be none.

4. Discussion: Authors state this inhibitor mainly affects collagen type 1, but during wound healing, first, collagen type 3 is synthesized, and collagen type 1 is produced much later. How does this affect the inhibitor?

5. Pig experiments: when did the treatment start for the dosing testing (experiment 2)?

Suppl fig 6: it seems that all treated wounds show more wound contraction than control wounds. Please comment. Was wound healing/contraction affected by the treatment?

Differences in scar quality in Suppl fig 7 seem rather small to me. Was an effect on rate of wound closure noted? If the cream was applied on open wounds, could a systemic effect occur that could influence also the other wounds?

Minor points:

Concentrations of in vivo experiments (max 1000 mg/kg/day) are not comparable to in vitro data (max 100 μ M range) versus % in cream (max 3%) : please also state the concentration in μ mole or nmol and/or comment on comparability of dosing in the different settings.

Fig 2: DAPI signal seems reduced in fig 2I versus 2G and H?

P17, line 2 from bottom: did not show any... instead of: did not show..

Reviewer #4 (Remarks to the Author):

To be sure, I am a medicinal chemistry professional researcher, and my specialty is to design and synthesize small molecules with biological activity. For this article, it is beyond

my professional scope. I can only talk about my shallow view: Overall, the study is deep and informative.

I suggest that the author provide the source of the compounds, the design basis of the compounds and the logic of science, rather than just the presentation of some data.

If the thinking and logic of a study are enlightening and reproducible for the future researchers, the significance of this study is undoubtedly significant.

Re: Topical application of an irreversible small molecule inhibitor of Lysyl Oxidases ameliorates skin scarring and fibrosis

Please find below point by point responses to the comments from the Reviewers:

Reviewer Comments

Reviewer #1 (Remarks to the Author)

"This descriptive study discloses the identification of a new LOX inhibitor. Currently, the project is preliminary, and controls are lacking. Models are not studied in depth

1. There is no description of "iLox" in the introduction. What is the IUPAC name of this, what is it exactly, how was it discovered? How specific is it? What is its MOA? Please provide actual data. Is it a copper chelator? Was this selected out of a combichem library with BAPN as an initial structure? Does it affect non-amine oxidases?"

We thank the Reviewer for their interest in the discovery of PXS-4787 (formerly described as iLOX). To provide the respective details we have added an additional Figure (Figure 1) to illustrate the compound design rationale and data driven structure optimisation leading to the discovery of new chemical entities PXS-4787 and PXS-6302. We have also included relevant mechanistic observations that highlight the unique properties of the compounds. The mode of action, distinct from copper chelation, is comprehensively described in Figure 2. Criteria for mechanism-based, irreversible inhibition (including time-dependency, substrate competition and jump dilution experiments) are fulfilled and evidence of on-target activity is provided by the biochemical activity as well as the *ex vivo* and *in vitro* inhibition displayed.

To offer an overview of the selectivity of PXS-6302 and 4787 we have added (in addition to the amine oxidase inhibition profile shown in Supp. Table 1, page 45) the results from screening against a broad panel of macromolecular targets (LeadProfilingScreen and SafetyScreen, Eurofins CEREP Panlabs; full results shown in Supp. Tables 2 and 5, pages 46 and 49, respectively).

"2. Figure 3 is confusing. Please separate bleomycin and wounding data.

Does inhibitor affect kinetics of wound closure in excisional punch biopsy model. Trichrome and anti- α -SMA staining would be helpful. Where is the untreated control? Please show primary data, not graphs."

We thank the Reviewer for highlighting these issues. Bleomycin data and wound healing data are now separated, with bleomycin fibrosis model data presented in Figure 6 for PXS-6302 and Supp. Figure 5 for PXS-4787 and wound healing data now presented in Figure 7 and Supp. Figure 6.

Massons trichrome (MT) images of bleomycin treated mice with PXS-6302 are included in Supp. Figure 6, including naïve, bleo+vehicle and bleo+PXS-6302 group images. MT images of wound healing in treated and control mice are now included in Supp. Figure 7.

Graphs have been changed from bar plots to dot plots or violin plots to show primary data (if $n < 10$ dot plots are used for $n > 10$ violin plots are used to show distribution).

Controls are included in revised figures – naïve animals in bleomycin group and vehicle and PXS-6302 treated groups where appropriate (composite skin score for example is not relevant for naïve animals for example and is therefore excluded from the panel).

Kinetics of wound closure: In murine studies we did not measure time to heal although we observed that application of PXS-4787 did appear to impact healing trajectory in the early phase (presumably through disruption of granulation tissue formation and stability). However all wounds had healed within

2 weeks in the murine model and the scars measured at 4 weeks post-injury were fully healed in all groups. Therefore any change to healing trajectory should not impact on the data presented here.

Our porcine studies were focused on application after initial healing to avoid potential issues of disruption to the healing process and better reflect likely clinical use. Our current clinical trial and expected application in further trials is for the treatment to be applied after healing is complete (re-epithelialisation).

In porcine studies application from 3 weeks after injury appeared most effective and given the likely interference with granulation tissue formation should LOX inhibition be applied early during wound repair we did not focus on healing trajectory but focused measures on scar tissue after completion of healing.

“3. As I understand the collagen crosslinking analysis was done on isolated fibroblasts, not tissue (methods). Please, explain Fig 3 D-H. What about crosslinking in bleomycin? Please provide a table outlining entire results of the proteomic analysis.”

Collagen crosslinking analysis was done on isolated fibroblasts (presented in Figure 4), and also in murine tissue samples from bleomycin fibrosis model (Fig. 6) and from excisional injury in mouse (from scar tissue from treated and control groups, presented in Figure 7 and Supp. Figure S7). This should now be clarified with the rearrangement of figures and separation of *in vitro*, bleomycin and wound healing models in the revised manuscript.

In all tests (*in vitro*, murine fibrosis and murine wound healing models) a reduction in cross-linking was observed.

Source data is now uploaded with the manuscript containing all crosslinking data from all models.

“4. Data to support effects on fibrosis are lacking eg real time PCR on key messages.”

We have now included additional data for the *in vivo* studies. This includes MT images of histology for both bleomycin and wound healing interventions in mice (Supp. Figure S6 and Supp. Figure 7). We have also included cross-linking analyses of *in vivo* studies for both murine fibrosis (bleomycin) and wound healing models (Figures 6, 7, and Supp. Figures 7 and 8). We did measure COL and LOX expression in response to the compound, and also used RNASeq to investigate the impact of the compound on the transcriptome of both fibroblasts and keratinocytes. We did not observe an effect on the transcriptome or RNA measured using these assays (Supp. Figures 2 and 3). As we expect the inhibitor to only target extracellular LOX and prevent collagen cross-linking, we do not expect RNA changes to be observed. This is potentially an advantage, as there appears to be no cellular response to the extracellular inhibition of LOX, suggesting it is unlikely there will be a compensatory response to treatment and also suggesting there is less chance of off-target effects on the main cells that will be exposed during topical treatment. This is now highlighted in the revised manuscript on page 12 and reproduced below;

...“Other studies have suggested LOX has non-enzymatic functions that are linked to epithelial differentiation (19, 20). Given that PXS-4787 would be applied topically, we investigated whether PXS-4787 had an impact on both keratinocyte and fibroblast transcriptomes using RNASeq. Since the target of interest is extracellular LOX activity, we wanted to understand if PXS-4787 also impacted on intracellular LOX and activity. 10 μ M PXS-4787 was applied to cultured fibroblasts and keratinocytes isolated from five different patients and treated for 24 hrs. RNASeq analysis showed only four genes with significant differential expression (FDR <0.05) in fibroblasts and two genes in keratinocytes (Supp. Figure S2 and Supp. Table S4). No change in COL1A1 or LOX RNA levels was observed indicating treatment does not impact COL1A1 or LOX transcription. The underlying biological mechanisms related to extracellular matrix biology of these six genes is unclear and these results suggest PXS-4787 does not impact on intracellular functions of LOX relevant for skin biology. This suggests application of the compound should effectively target extracellular LOX without impacting on skin cell phenotype or leading to a compensatory response.”

Reviewer #2 (Remarks to the Author)

“A few (minor) suggestions are made to increase clarity of the manuscript. Questions should be addressed regarding whether or not elastin cross-linking was examined, and whether or not collagen fibril diameters were or could be analyzed from EM images. Parameters of MS/MS analyses and data would have been helpful to have been provided as well.”

Elastin cross-links were not examined in this study – our assay only measured collagen cross-linking. This is now acknowledged in the limitations part of the discussion on page 27.

...“We have also not measured the impact on elastin and elastin cross-links, which may contribute significantly to changes in scar tissue.”

MS/MS parameters are now included with LLOQ table included as supp. Table 6 for each cross-linking assay reported and we have also referenced our previous work detailing the analyses of collagen cross-linking (ref# 34 in revised manuscript).

We have not measured collagen fibril diameters from the EM images we obtained.

“Specific comments are provided below for the authors to consider.

1. Figure 2: The legend should indicate that cell layers were used for cross-link analyses. It is unclear which collagen propeptides were measured, and in media or cell layer, and what methodological approach was used. This is explained in supplementary information, but brief statements would help clarify these basic parameters in the main article.”

These details have been included in brief in the revised figure legend and associated main text (for Fig 4 in revised manuscript). We have also separated figures to demonstrate clearly we have measured cross-linking *in vitro* and in both murine models of healing and of fibrosis (Figures 6, 7, and Supp. Figure 7).

“2. “Coherency analysis” is not a very commonly used term and is not clarified in the main text. Perhaps consider a different way to describe quantitative analyses of immunofluorescence images.”

Coherency provides an opportunity to look at the alignment of fibres rather than just quantity. Scar tissue commonly has a higher score due to parallel fibre arrangements in contrast to the more random fibre alignments found in normal skin. We have clarified this analytical approach in the manuscript to present clearly what this approach provides, page 12.

“Furthermore, coherency analysis (18) was used to determine collagen alignment. This analysis specifically analyses the alignment of collagen fibres, with more alignment increasing the score and more random distribution provided a lower score. In scar tissue scores are higher as fibres are densely parallel aligned in contrast to normal skin (18). In both treatment groups PXS-4787 significantly reduced coherency ($p < 0.01$) when compared to control (Figure 4K) suggesting inhibition of cross-linking reduces the stability of deposited collagen and changes the distribution to be more like that seen in normal skin *in vitro*.”

“3. Figure 4A control group: The legend should indicate that the time points refer to when iLOX treatment began, and the common 12 week time point of harvest of the tissues should also be indicated in the legend. Otherwise it is unclear and confusing regarding how the single control relates to the time points of the experimental samples.”

We thank the Reviewer for pointing this out and have taken the feedback on board. This has now been clarified in the revised figure (now Supp. Figures 8-10). All figure legends have been amended to reflect that the time point of 1, 2 or 3 weeks refers to commencement post-injury not duration of treatment.

“4. EM images and analyses: Collagen cross-sections to measure diameters of fibrils would be of interest to determine if these are the same or different in inhibitor treated skin. Collagen diameters often increase as a function of LOX inhibitors.”

We have not been able to process additional samples for measuring fibril diameter but agree this will be of interest with respect to the impact of the inhibitors. We are currently conducting Phase I trials of

PXS-6302 and may have an opportunity to investigate this with biopsy samples collected (ACTRN12621001545853).

"5. Discussion: At the bottom of page 18 "This reduction in cross-links will reduce collagen fibril strength (32) and tissue stiffness (33)", please change "will" to "may" since no data regarding fibril strength was presented in this manuscript."

We agree with the Reviewer and have amended the text as suggested.

"6. Methods: Crosslink analyses: Description of methods stopped at HPLC step. MS/MS conditions and data analyses strategies used should also be presented, or at least relevant respective publications cited."

The description has been amended and reference added with complete methodology (34). The lower limit of detection is also included in the supplementary information (Table S6).

"7. Skin contains elastin in addition to collagen. Were desmosine and isodesmosine elastin cross-links analyzed by MS/MS, which are also dependent on LOX and can contribute to fibrosis?"

Elastin cross-links have not been measured in these studies as our approach using MS/MS was validated for collagen cross-links for these studies. As acknowledged above (response to reviewer 2 overview) this is a key limitation and has been included in the revised discussion.

"8. Reference 14 and 15 are identical."

We thank the Reviewer for pointing this out and have corrected the issue.

"9. Consider using PMID: 11779117 instead of reference 39."

We agree with the Reviewer and have amended the reference as suggested (now reference 35 in revised manuscript).

Reviewer #3 (Remarks to the Author)

"This paper describes in vitro and in vivo results of testing an anti-fibrotic agent based on inhibition of lysyl oxidases. Penetration on human skin, toxicology data and effects in reducing collagen crosslinking and reduced scarring after skin injury in different animal models are described. The data are well described and extensive sets of experiments are presented. Some experimental details need to be added, but especially, the following points need to be addressed before this paper can be published.

1. Bleomycin model: Fig 3. Describes Bleomycin induced fibrosis. However, bleomycin is also used to reduce fibroblast proliferation eg in keloid scars. How effective was this to induce fibrosis?

Treatment with the inhibitors was started already at day 3: by that time, there will not be a fully developed hypertrophic or fibrotic skin. Why was the treatment started this early and not after 20 days? Were the slides of this experiment scored blinded to treatment?

Please also show histology & crosslink measures of control and bleomycin versus bleomycin iLOX treated mouse skin."

Bleomycin is used clinically for treatment of keloid but has also been extensively used as a murine model of fibrosis across multiple tissues including skin and lung. We have included references for the protocol used here and for its use to induce skin fibrosis (references 21,22). Bleomycin is the most widely used model of fibrosis in the mouse despite acknowledged limitations of this model with respect to human disease.

The treatment was commenced to investigate whether LOX inhibition could be effective in reducing collagen cross-linking in vivo and whether this may impact on the trajectory of fibrosis in this model. Since LOX inhibition will only impact on new cross-linking this model of early treatment allowed earlier detection of changes in fibrosis. This is supported by the significant changes in immature cross-links observed in this model (Fig. 6).

Slide assessment was performed by blinded assessors. This has been clarified in manuscript (Supp. Methods, Bleomycin injury).

Histology images have been included (Supp. Figure 6) for bleomycin treated groups.

Crosslink analysis was conducted in vitro using isolated human fibroblasts, in murine wound model (excision injury) and in the murine bleomycin model and these data are now included in the revised manuscript (Figures 6, 7 Supp. Figure 7)

“2. Full thickness wounds in mice: Please present these data in a separate figure to avoid mix-up with the bleomycin experiment.

Was wound healing delayed in alignment with lower collagen production? Was wound contraction affected? Were there effects on skin strength noted? What was the duration of the treatment? Were control wounds treated with the vehicle?

From the description it seems that the iLOX was administered on the open wounds. In that case, it would be relevant to determine the systemic load of the inhibitor, as it is likely to be higher than if penetrated through intact skin.”

We thank the Reviewer for highlighting a potential source of confusion and have made the suggested change in revised manuscript (Figures 6, 7 and Supp. Figure 7).

We did not measure time to heal in the murine studies although we observed that application of PXS-4787 did appear to impact on healing trajectory in the acute stage after injury. This would be expected through an impact on the deposition and stability of granulation tissue that is rich in newly synthesised collagen. However all our studies focused on fully healed scars in both murine and porcine models. Our porcine work focused on administering PXS-6302 after re-epithelialisation as this more closely mimics what we plan for human clinical trials and is a key potential benefit of the LOX target as it is effective at late stages of healing (Supp. Fig. S8-S10). Therefore the data is not impacted by changes in healing time as all studies were conducted on fully healed scar.

No effects were observed on contraction in the pig model. Tensile strength was tested in larger scars created using a burn wound model in the pig (Figure 8). No reduction in scar strength was observed, although a trend for increased pliability was seen. Duration of treatment for both murine and porcine models have now been clarified in the manuscript.

All control wounds are treated with vehicle – this is clarified in the manuscript.

Systemic load of inhibitor. We have data demonstrating in a rat model provided 1000mg/kg/day for one week systemic exposure was only 0.4 μ M. In our porcine models we delivered approximately 500mg of cream per injury site. This would provide a maximum of 3g of 3% PXS-4787 cream applied per day, or 90mg of PXS-4787. The pigs were a minimum of 30kg by the time of initial treatment, providing a maximum daily dose of 3mg/kg/day, 300-fold less than that in the rat model and insufficient for any systemic effects on wounds to which the cream was not applied or for other systemic impacts. This is a key benefit of topical administration since it provides effective dosing at the site with minimal potential systemic toxicity due to minimal dose required.

“3. Has a validated scar assessment technique been used for the scar evaluation? Just a scoring system of better vs worse does not seem adequate; each scar should be evaluated at its own merit, not in a comparative fashion to the other treatment groups. The latter method could artificially increase differences due to the fact that raters are asked to establish a ranking where there might be none.”

Blinded assessment of photos is a useful tool for scar assessment as it is also commonly used clinically. These scars were assessed by at least 3 plastic surgeons independently and blinded to treatment. The surgeons have now scored each scar independently (not comparative) as well as ranked (comparative)

scoring. Both analyses show improved scar and are included in the revised manuscript (Fig. 8 and Fig. S8). Furthermore, the physical properties of the scar (tensile strength and pliability) were also assessed as part of the porcine burn model. Results indicated no loss of tissue strength, with 5/6 treated scars showing greater pliability than their vehicle treated control.

“4. Discussion: Authors state this inhibitor mainly affects collagen type 1, but during wound healing, first, collagen type 3 is synthesized, and collagen type 1 is produced much later. How does this affect the inhibitor?”

The application of the inhibitor in experimental studies may have impacted on collagen III deposition but this was not measured. Porcine studies and our clinical application of these compounds is targeted towards latter stages of healing (post re-epithelialisation and in particular during remodelling) and is therefore targeted at Collagen I deposition and stability rather than Collagen III that is produced much earlier. Our studies particularly in the porcine model reflect this potential use with application commencing at 3 weeks post injury being most effective (figure S8) and administration commencing after complete re-epithelialisation effective in both burn and excision porcine injury models (Fig 8).

“5. Pig experiments: when did the treatment start for the dosing testing (experiment 2)?”

“Suppl fig 6: it seems that all treated wounds show more wound contraction than control wounds. Please comment. Was wound healing/contraction affected by the treatment? Differences in scar quality in Suppl fig 7 seem rather small to me. Was an effect on rate of wound closure noted? If the cream was applied on open wounds, could a systemic effect occur that could influence also the other wounds?”

For the dose response, treatment was commenced at time of re-epithelialisation of the wounds, approximately 2 weeks after injury. This has now been clarified in the manuscript.

For the porcine dose response, the cream is not applied to open wounds but in murine models as it was applied early this may have impacted on healing trajectory. However the dose provided was a maximum of 3mg/kg/day in our porcine studies, which is over 300-fold less than a dose we have studied in rat (daily for 1 week, 1000mg/kg/day) and which resulted in negligible systemic levels of PXS-4787 or PXS-6302. Therefore we are confident there is no possibility of a systemic effect because the topical dose provided is too low to contribute to a systemic change.

“Minor points: Concentrations of in vivo experiments (max 1000 mg/kg/day) are not comparable to in vitro data (max 100 µM range) versus % in cream (max 3%) : please also state the concentration in µmole or nmol and/or comment on comparability of dosing in the different settings.”

The use of micromolar concentrations in solution in our in vitro studies was to provide proof-of-principle for effective inhibition of LOX in vitro. The use of topical cream at percentages up to 3% was to closely mimic expected clinical application. For porcine studies approx. 500mg cream was applied to each wound, providing a maximum daily dose in the animals treated with PXS-4787 of 30mg/kg/day (approximately since pigs gain weight rapidly this would be maximal dose at time dosing commenced). Studies using up to 1000mg/kg/day were focused on toxicity and pharmacokinetics to provide evidence for safety of application and were far in excess of clinical application or that used in our porcine models.

“Fig 2: DAPI signal seems reduced in fig 2I versus 2G and H? P17, line 2 from bottom: did not show any... instead of: did not any show.”

The brightness for this image has been increased with no other changes to more clearly show the DAPI staining in this panel (Figure 4 panel H in revised manuscript).

Typographic change has been corrected and we thank the Reviewer for highlighting these.

Reviewer #4 (Remarks to the Author)

“To be sure, I am a medicinal chemistry professional researcher, and my specialty is to design and synthesize small molecules with biological activity. For this article, it is beyond my professional scope. I can only talk about my shallow view: Overall, the study is deep and informative. I suggest that the author provide the source of the compounds, the design basis of the compounds and the logic of science, rather than just the presentation of some data. If the thinking and logic of a study are enlightening and reproducible for the future researchers, the significance of this study is undoubtedly significant.”

We thank the Reviewer for the positive feedback on the significance of our studies and interest in the discovery of PXS-4787 (formerly iLOX). We have taken the opportunity to include an additional Figure (Figure 1) to illustrate the compound design rationale and data driven structure optimisation leading to the discovery of new chemical entities PXS-4787 and PXS-6302. We have also included relevant mechanistic observations that highlight the unique properties of the compounds. The mode of action, distinct from copper chelation, is comprehensively described in Figure 2. Criteria for mechanism-based, irreversible inhibition (including time-dependency, substrate competition and jump dilution experiments) are fulfilled and evidence of on-target activity is provided by the biochemical activity as well as the *ex vivo* and *in vitro* inhibition displayed.

To provide an overview of the selectivity of PXS-6302 and 4787 we have added (in addition to the amine oxidase inhibition profile shown in Supp. Table 1, page 45) the results from screening against a broad panel of macromolecular targets (LeadProfilingScreen and SafetyScreen, Eurofins CEREP Panlabs; full results shown in Supp. Tables 2 and 5, pages 46 and 49, respectively).”

REVIEWERS' COMMENTS

Reviewer #1 (Remarks to the Author):

The revisions are extremely helpful and provide an overall context for the study.

Reviewer #2 (Remarks to the Author):

The revised manuscript now documenting the development and use of a novel lysyl oxidase inhibitor to address skin fibrosis and keloid development has been improved with direct responses to the previous reviews, and modifications to the manuscript itself. I have no further comments for the authors to address.

Reviewer #3 (Remarks to the Author):

The findings reported here are potentially of great value for developing therapies against various forms of fibrosis.

Authors have improved the manuscript considerably after the first review process. Some minor points remain:

Abstract: reduction in collagen fibril thickness is mentioned here. The data on collagen fiber thickness are somewhat hidden in supplementary data fig 8C, and only show a minimal effect. If authors consider this an important finding, the data should be in the main body of the manuscript. However, given the minor effect, it would be better to remove this statement from the abstract.

Fig 4: How do authors explain the large variation in hydroxyproline and crosslinks in fig 4 A – E?

Reviewer #4 (Remarks to the Author):

I am very satisfied with the author's reply and the new revised version.

Re: Topical application of an irreversible small molecule inhibitor of Lysyl Oxidases ameliorates skin scarring and fibrosis

Please find below point by point responses to the comments from the Reviewers:

Reviewer Comments

Reviewer #3 (Remarks to the Author):

The findings reported here are potentially of great value for developing therapies against various forms of fibrosis.

Authors have improved the manuscript considerably after the first review process. Some minor points remain:

Abstract: reduction in collagen fibril thickness is mentioned here. The data on collagen fiber thickness are somewhat hidden in supplementary data fig 8C, and only show a minimal effect. If authors consider this an important finding, the data should be in the main body of the manuscript. However, given the minor effect, it would be better to remove this statement from the abstract.

The authors agree with the reviewer that this is not a major finding and have removed the statement from the final abstract.

Fig 4: How do authors explain the large variation in hydroxyproline and crosslinks in fig 4 A – E?

Primary human fibroblasts were used for these experiments. These cells were from six different participants (or three different participants for immunohistochemistry) and therefore had a high level of variation in their baseline production of collagen and cross-linking. The datapoints show biological replicates not technical replicates (this is stated in figure legend). The variation is likely due to differences in age, gender, body site from which primary cells were obtained. In all cases the effect of the inhibitor was the same with respect to reduction in cross-linking observed (and paired analysis was used for this to compare the reduction in each individual patient sample).

The use of biological replicates as the reason for variation in this part of the study has been clarified in the manuscript.